# Multi-Feature Transformer-Based Learning for Continuous Human Motion Recognition with High Similarity Using mmWave FMCW Radar

**DOI:** 10.3390/s22218409

**Published:** 2022-11-01

**Authors:** Yuh-Shyan Chen, Kuang-Hung Cheng, You-An Xu, Tong-Ying Juang

**Affiliations:** Department of Computer Science and Information Engineering, National Taipei University, No. 151, University Rd., San Shia District, New Taipei City 23741, Taiwan

**Keywords:** FMCW, continuous human motion recognition, high similarity, multi-feature, Transformer

## Abstract

Doppler-radar-based continuous human motion recognition recently has attracted extensive attention, which is a favorable choice for privacy and personal security. Existing results of continuous human motion recognition (CHMR) using mmWave FMCW Radar are not considered the continuous human motion with the high similarity problem. In this paper, we proposed a new CHMR algorithm with the consideration of the high similarity (HS) problem, called as CHMR-HS, by using the modified Transformer-based learning model. As far as we know, this is the first result in the literature to investigate the continuous HMR with the high similarity. To obtain the clear FMCW radar images, the background and target signals of the detected human are separated through the background denoising and the target extraction algorithms. To investigate the effects of the spectral-temporal multi-features with different dimensions, Doppler, range, and angle signatures are extracted as the 2D features and range-Doppler-time and range-angle-time signatures are extracted as the 3D features. The 2D/3D features are trained into the adjusted Transformer-encoder model to distinguish the difference of the high-similarity human motions. The conventional Transformer-decoder model is also re-designed to be Transformer-sequential-decoder model such that Transformer-sequential-decoder model can successfully recognize the continuous human motions with the high similarity. The experimental results show that the accuracy of our proposed CHMR-HS scheme are 95.2% and 94.5% if using 3D and 2D features, the simulation results also illustrates that our CHMR-HS scheme has advantages over existing CHMR schemes.

## 1. Introduction

Human Motion Recognition (HMR) has recently attracted extensive attention [1,2]. For instance, HMR can detect or prevent falls in residential care homes for elderly people. Human motion recognition (HMR) can be achieved using wearable-sensor-based technologies, camera-based technologies, and radar-based technologies, but wearable-sensor-based solutions require users to attach multiple sensors on their bodies, resulting in a great inconvenience, and camera-based technologies raise privacy concerns [1]. Due to the impact of the COVID-19 disease, most people are increasingly accepting the use of contactless radar-based methods to record human behaviors under privacy-preserving conditions, compared to wearable-sensor-based technologies and camera-based technologies. The key advantages of mmWave FMCW radar, compared with LiDar, are that it is not affected by light, and it is of low cost. Due to these characteristics, it is very useful to apply mmWave FMCW radar applications for automatic car driving and home care, especially for night care for the elderly.

The frequency modulated continuous wave (FMCW) radar-based solutions have achieved many results in human motion recognition (HMR) results [3,4]. In the early stage, radar-based HMR results are mainly focused on a single HMR for an observation that just only includes one type of motion with fixed time duration from mmWave FMCW radar sensors. Conventional Doppler-radar-based HMR methods [1] mainly utilize the FMCW spectrogram generated using a short-time Fourier transform (STFT) to extract handcrafted features, and machine learning or deep learning classifier is utilized for the feature-based classification. For instance, these include the Doppler-radar-based classification of pedestrians, bicycle users, and human daily activities [5,6,7]. Machine learning or deep learning-based human motion recognition results recently are widely adopted to improve the recognition accuracy. For instance, the recognition of a human body’s gait using Convolutional Neural Networks (CNN) is investigated in [8,9].

In the normal human activities, people usually perform motions one after the other for varying durations [1]. More recently, continuous HMR based on micro-Doppler features from mmWave FMCW radar has become the focus of research [1]. Some continuous HMR methods mainly rely on individual recurrent neural network or sliding-window-based approaches, which cannot effectively exploit all the temporal information to predict motions. Zhao et al. [1] then proposed an end-to-end network for continuous motion recognition via radar radios. Zhang et al. [2] proposed a dynamic continuous hand gesture recognition based on a frequency-modulated continuous wave radar sensor. Shrestha et al. [10] proposed a continuous human activity monitoring and classification from FMCW radar based on recurrent LSTM and Bi-LSTM network architectures. Zhao et al. [11] also proposed a continuous human motion recognition using micro-Doppler signatures in the scenario with micro-motion interference, using Empirical Mode Decomposition (EMD) to remove non-target interference.

However, all of the mentioned existing continuous HMR results do not investigate the classification issue for the high-similarity motions. We consider three different high-similarity motion groups; walking group, running group, and vertical-motion group. The walking group includes motions of walking forward, walking backward, walking right, and walking left. The running group includes motions of running forward, running backward, running right, and running left. The vertical-motion group contains motions of lying down and falling. We observe that if the continuous HMR for the observed motions are from the same high-similarity motion group, the lower the recognition accuracy will be. This is because that the similar micro-Doppler signature is obtained for all motions selected from the same high-similarity motion group. Therefore, the main motivation and contribution of this work is to propose a new continuous HMR (CHMR) algorithm with the consideration of the high similarity (HS) problem, which is termed the CHMR-HS scheme in this paper.

For HMR and continuous HMR, the Doppler signature and the Range-Doppler-Time signature from FMCW radar are the most common extracted features in the existing literature [10,11,12,13]. Papanastasiou et al. [12] proposed a deep learning-based identification of human gait using the Range-Doppler-Time signature. Shrestha et al. [10] proposed a continuous human motion classification from the FMCW radar with Bi-LSTM networks using Doppler signature spectrograms. Zhao et al. [11] proposed a continuous human motion recognition using the Doppler signature. Apart from the use of Doppler-based signature spectrograms, some new recent works further consider the three-dimensional radar data cube; that is, the Range-Doppler-Angle, for the HMR. For instance, Zheng et al. [13] proposed a dynamic hand gesture recognition algorithm by extracting the features of the Range-Doppler Map (RDM), Range-Azimuth Map (RAM), and Range-Elevation Map (REM) of time signatures as an input of the trained model.

In this paper, we propose a continuous HMR algorithm with high similarity (CHMR-HS) by using multi-features. To investigate the effects of the spectral–temporal multi-features with different dimensions, Doppler, range, and angle signatures are extracted as the 2D features, and range-Doppler-time and range-angle-time signatures are extracted as the 3D features. The 2D/3D features are separately trained into the modified Transformer-encoder model to effectively distinguish the difference of the high-similarity human motions. Our CHMR-HR scheme is to re-design a signature-to-signature learning model, where the input-signature are the 2D/3D feature signatures and the output-signature is the recognition result of the motions one after the other for varying durations. It is also worth noting that the 2D/3D feature signatures with multiple motions are fed into our designed signature-to-signature learning model to produce the corresponding multiple motion recognition results under the non-fixed recognition number.

In the sequence-to-sequence learning model, we propose a Transformer-based learning model, which is constructed using modified Transformer-encoder and Transformer-decoder models. In our Transformer-encoder part, the Doppler, angle, and range feature signatures are independently embedded and decoded using disjoint Transformer-encoders, and the corresponding output features are then fused to the fused feature. The conventional Transformer-decoder model is re-designed to be the Transformer-sequential-decoder model, such that the Transformer-sequential-decoder model can recognize continuous human motion with the high similarity from the trained Transformer-encoder model. In our Transformer-sequential-decoder part, an extra weight is trained sequentially via the order of the Doppler feature, angle feature, and range feature, with their cross-attentions from Transformer-encoder model. The trained extra weight will be added into the self-attention to enhance our signature-to-signature learning model. Furthermore, our Transformer-sequential-decoder is re-designed to allow our continuous HMR scheme to recognize the non-fixed number of motions with the dynamic motion duration. It is noted that connectionist temporal classification (CTC) [14] technique is widely used to perform the model training between blank segments and motion labels, due to the CTC classifiers being able to effectively predict the motions between motions in the early training stage. In our work, the CTC technique [14] is also tailored into our trained model, so that the early training stage of the CHMR-HS scheme can converge faster. As shown in Figure 1, our CHMR-HS scheme is illustrated. The contributions of this paper are listed as follows.
As far as we know, this is the first result in the literature to investigate continuous HMR with the consideration of high-similarity motions. It is observed that if the observed continuous motions are from the same high-similarity motion group, the recognition accuracy is degraded due to the similar micro-Doppler signature being obtained for the motions selected from the same high-similarity motion group.To investigate the effects of the spectral–temporal multi-features with different dimensions, our proposed CHMR-HS scheme is trained using 2D/3D multi-features, where Doppler, range, and angle signatures are extracted as the 2D features, and the range-Doppler-time and range-angle-time signatures are extracted as the 3D features.To effectively recognize continuous human motion recognition with high similarity, our CHMR-HS scheme presents a new Transformer-based learning model, where the new Transformer-based learning model is constructed via modified Transformer-encoder and the new proposed Transformer-sequential-decoder models.The recognition accuracies of our proposed CHMR-HS scheme for continuous human motion recognition are 95.2% and 94.5% using 3D and 2D features under the motion samples of CHMR that are randomly selected from any motion groups, which are higher than those of the existing results by 2.1% and 1.4%, respectively.The recognition accuracies of our proposed CHMR-HS scheme are 94.5% and 94.1% using 3D and 2D features, which are higher than those of the existing results by 12% and 11.6%, if motion samples of CHMR are only selected from the same high-similarly motion groups. It is shown that our proposed CHMR-HS scheme for CHMR can more effectively improve the recognition accuracy for CHMR with high similarity.

The rest of the paper is organized as follows. Section 2 describes the related work and motivation. Section 3 introduces the system model, problem formulation, and basic ideas. Section 4 describes the proposed CHMR-HS scheme. Section 5 discusses the performance results. Finally, Section 6 concludes the paper.

## 2. Related Work

This section first describes the related works in Section 2.1 and discusses the research motivation for the study in Section 2.2.

### 2.1. Related Work

The time-frequency diagram of FMCW radar is adopted for HMR, and the phase difference of the radar echo data is extracted using Fourier transform. Some related works for micro-Doppler modulation [15], background subtraction [16], and interference mitigation [17] are described. Chen et al. [15] proposed a micro-Doppler modulation model to study the vibration and rotation from the micro-motions of the detected objects, and to analyze the micro-Doppler effects. The moving target indication (MIT) digital filtering method is designed by Ash et al. [16] as the background subtraction to effectively remove returns from stationary and slow-moving objects. Constant false alarm rate (CFAR) detector-based approaches are proposed by Wang [17] for the interference mitigation of FMCW radars.

There are some interesting single HMR works based on machine learning or deep learning algorithms. Kim et al. [18] proposed a human detection and activity classification for single motion based on micro-Doppler signatures using deep convolutional neural networks. To alleviate the burden of collecting and annotating a large number of radar samples, Li et al. [8] proposed an HMR with an instance-based transfer learning (ITL) method under the limited radar micro-Doppler (MD) signatures. Sakagami et al. [9] proposed an accuracy improvement of HMR using CNN, using three 2D domains (Time-Doppler, Time-Range, and Range-Doppler) of mmWave FMCW as the motion data. Khalid et al. [19] proposed a multi-view CNN-LSTM architecture for the radar-based HMR, while the multi-view radar features includes raw features, energy dispersion based features, temporal difference based features, and auxiliary features.

Continuous HMR results are more practical, due to the human usually performing motions one after the other for varying durations. So far, there are some useful HMR results with the assistance of machine learning or deep learning algorithms. Shrestha et al. [10] proposed a continuous human action classification from FMCW radar with Bi-LSTM networks, while this work utilizes radar data in the form of a continuous temporal sequence of micro-Doppler or range-time information. Zhao et al. [1] then proposed an end-to-end network for continuous motion recognition via radar radios. To guarantee the effective use of all the temporal information, the attention-based encoder–decoder structure encodes the fused data and selects the temporal information for recognition. Zhang et al. [2] proposed a dynamic continuous hand gesture recognition based on a frequency-modulated continuous wave radar sensor, and a recurrent 3D convolutional neural network is used to perform the classification of dynamic hand gestures. Distributed radar-based solutions and range-Doppler trajectory methods are recently further considered for CHMR. Guendel et al. [20] proposed a continuous Activities of Daily Living (ADL) recognition in an arbitrary movement direction using distributed pulsed Ultra-Wideband (UWB) radars in a coordinated network. To recognize continuous human motions with various conditions for the real-living environment, Ding et al. [21] proposed a novel dynamic range-Doppler trajectory (DRDT) method using the FMCW radar system.

Transformer-based encoder–decoder architecture is very suitable for the training the time-series data for the continuous HMR problem. Vaswani et al. [22] initially proposed a Transformer-based encoder–decoder architecture with the self-attention mechanism, while the self-attention mechanism of the Transformer-based model is very efficient for the sequence-to-sequence learning task. Hu et al. [23] modified the Transformer-based architecture by proposing a multimodal multitask learning with a unified Transformer. Lohrenz et al. [24] proposed a multi-encoder learning and stream fusion for Transformer-based end-to-end automatic speech recognition. It is valuable to develop the new continuous HMR scheme based on Transformer-based encoder–decoder architecture.

### 2.2. Motivation

Radar-based HMR technology can provide people with human motion monitoring services with high stability, high privacy, and more privacy. All of the mentioned existing CHMR results do not investigate the classification issue for the high-similarity motions. For instance, some CHMR results [1,2,11] utilized MLP classifier, CTC classifier, and RNN-based classifier to recognize the continuous human motion. In the paper, three different high-similarity motion groups, the walking group, running group, and vertical-motion group, are considered in this paper. The recognition of continuous human motion with high similarity has high challenges under the limitation of only radar-based images being allowed used to satisfy the purpose of the high stability, high privacy, and more privacy.

As mentioned before, the walking group includes motions of walking forward, walking backward, walking right, and walking left; the running group includes motions of running forward, running backward, running right, and running left; and the vertical-motion group contains motions of lying down and falling. It is observed that if the continuous HMR for the observed motions is from the same high-similarity motion group, the recognition accuracy is degraded. For example, as shown in Figure 2a, there are four motions. In order, they are running forward, running forward, running right, and running left from the same running group. It can be observed that each Doppler signature is very similar with each other. In contrast, as shown in Figure 2b, there are still four motions. In order, they are walking forward, falling, walking backward, and lying down. The motions of walking forward and walking backward belong to the walking group, but falling and lying down belong to the vertical-motion group. The Doppler signatures of walking forward and walking backward are similar, and the Doppler signatures of falling and lying are similar. However, the Doppler signatures of walking forward or walking backward with the Doppler signatures of falling and lying are quite different. If the Doppler signatures of the observed period are quite different, it will be more easy for the CHMR to recognize the different motions. However, if the all the Doppler signatures of the observed period are similar, it will increase the difficulty of CHMR and degrade the recognition accuracy for continuous human motions. Therefore, the main motivation of this work is to propose a new CHMR algorithm with the consideration of the high similarity (HS) issue, called the CHMR-HS scheme in this paper.

## 3. Preliminaries

This section describes the system model, the problem formulation, and the basic idea in Section 3.1, Section 3.2 and Section 3.3, respectively.

### 3.1. System Model

The system model of the CHMR-HS scheme consists of the radar data collection module, the raw data preprocessing module, and the human motion recognition module. The radar data collection module is to collect the raw frequency-modulated continuous wave (FMCW) radar, the raw data preprocessing module is to extract 2D/3D features from the raw FMCW data, and the human motion recognition module is to perform the continuous HMR with HS capability.

Twelve motion patterns are considered in this paper, including waving, jumping forward, falling, lying down, walking forward, walking backward, walking left, walking right, running forward, running backward, running left, and running right. Among them, the motions of walking forward, walking backward, walking right, and walking left with similar micro-Doppler effects are collected as the walking group. The motions of running forward, running backward, running right, and running left with similar micro-Doppler effects are collected as the running group. The motions of lying down and falling with similar micro-Doppler effects are collected as the vertical-motion group.

In the radar data collection module, as shown in Figure 3, the raw data are extracted from the AWR1642-BOOST mmWave FMCW sensing device from Texas Instruments [25], which is the IWR1642 single-chip from a 76 GHz to 81 GHz mmWave sensor integrating DSP and microcontroller (MCU) evaluation module. The basic principle of the mmWave FMCW sensing device is to use a frequency-modulated continuous wave (FMCW) for frequencies emitted at different times via MIMO; the distance between objects is detected through the difference between the transmitting antenna and the receiving antenna, and the Doppler frequency shift is calculated through the phase difference in the sine wave. The criteria for designing the FMCW Radar is that FMCW Radars transmit a frequency modulated continuous wave signal, and a reflected FMCW signal from the remote target is mixed with the transmitted signal to generate a signal at an intermediate frequency (IF), which is used in the range, Doppler, and angle calculations [1,2].

The raw data preprocessing module, as shown in Figure 3, is to eliminate the noise and interference problems, to provide more easily recognized FMCW radar images to the human motion recognition module. The main work is to perform a background removal operation, which results in the cross-reflection effects of non-dynamic objects due to the obfuscation of detected objects and the background. Another major task is to further perform the feature enhancement operation to enhance the features of high-movement observed human motions through the Doppler frequency shift of the highly reflective objects. After the background removal and feature enhancement operations, the 2D/3D features will be extracted.

In the human motion recognition module, as shown in Figure 3, the raw 2D/3D features are separately trained and fused together to be fused features in the Transformer-encoder model. The importance of the 2D/3D features of each time segment of continuous human motion will be trained in the Transformer-sequential-decoder model, as shown in Figure 3. A joint decoding is used for the recognition optimization.

### 3.2. Problem Formulation

Before defining the problem formulation of our work, some notations are defined. Let *S* be a set of training samples of radar images, $S=x_{1},\ldots ,x_{i},\ldots ,x_{I}$, where 1≤i≤I, $x_{i}$ is *i*-the training sample of radar images from continuous human motions for a period of time, and let *I* be the total training samples. Note that $x_{i}$ can be a sample of 2D radar features or 3D radar features for a period of time; let $S_{2D}$ be simplified as *S* if all $x_{i}\in S$ are 2D radar features for a period of time, where each $x_{i}$ contains *i*-th training sample of the Doppler signature, angle signature (including elevation or azimuth), and range signature for a period of time. In addition, we may let $S_{3D}$ be also simplified as *S* if all $x_{i}\in S$ are 3D radar features for a period of time, where each $x_{i}$ contains the *i*-th training sample of the range-Doppler signature and range-angle signature for a period of time. In this work, we define *K* different motions and *n* high-similarity motion groups, such that *m* different motions are distributed into *n* high-similarity motion groups. For $x_{i}\in S$, for $S\in S_{2D}$ or $S_{3D}$, each continuous human motion contains *J* motions, their corresponding labels of *J* motions are denoted as label sequence $\left(l_{i}^{1},\ldots ,l_{i}^{j},\ldots ,l_{i}^{J}\right)$ for $x_{i}$, where 1≤j≤J. Let $Y=y_{1},\ldots ,y_{i},\ldots ,y_{I}$ be the set of label sequences for *S*, if $S\in S_{2D}$ or $S_{3D}$, where the label sequence of $x_{i}$ is $y_{i}=\left(l_{i}^{1},\ldots ,l_{i}^{j},\ldots ,l_{i}^{J}\right)$ and *J* is a dynamic value for different $x_{i}$.

Our work has three decoders, there are the CTC decoder, Transformer-sequential-decode, and Transformer-importance-decoder. We first describe the loss functions of these three decoders as follows.

First, the loss function of the CTC decoder, $L_{CTC}$, is described below. To overcome the blank segment and the meaningless feature sequence for the long-time series motion recognition, the connectionist temporal classification (CTC) [14] technique is useful for our scheme, so the CTC technique is integrated into our scheme as one of our feature decoders. Given a set of training samples of radar images, $S=(x_{1},...,x_{i},\ldots ,x_{I})$, where 1≤i≤I, and $Y=(y_{1},\ldots y_{i}\ldots ,y_{I})$ is a set of label sequences for *S*, if $S\in S_{2D}$ or $S_{3D}$, where the label sequence of $x_{i}$ is $y_{i}=\left(l_{i}^{1},\ldots ,l_{i}^{j},\ldots ,l_{i}^{J}\right)$, and $l_{i}^{j}$ represents the label of the *j*-th motion representation for $x_{i}$, where 1≤j≤J and *J* is a dynamic value for different $x_{i}$. Let *i*-th continuous human motions $x_{i}$ of *S* have the label sequence $\left(l_{i}^{1},\ldots ,l_{i}^{j},\ldots ,l_{i}^{J}\right)$, if it is assumed that the predicted result classified by the CTC decoder of $x_{i}$ is ${\hat{y}}_{i}=\left({\hat{l}}_{i}^{1},\ldots ,{\hat{l}}_{i}^{j},\ldots ,{\hat{l}}_{i}^{f}\right)$, where ${\hat{y}}_{i}=P\left(y_{i}|x_{i}\right)$ under $P(y_{i}|x_{i})$ is probability of hypothesis $y_{i}$. We have a set of predicted label sequences $\hat{Y}=\left({\hat{y}}_{1},\ldots ,{\hat{y}}_{i},\ldots ,{\hat{y}}_{I}\right)$ for $S=(x_{1},...,x_{i},\ldots ,x_{I})$. This work aims to find the best solution for the classification of the prediction results of the CTC decoder. Let ${\hat{Y}}^{*}=arg\;max\;P\left(Y|S\right)$, where ${\hat{Y}}^{*}=\left({\hat{y}}_{1}^{*},\ldots ,{\hat{y}}_{i}^{*},\ldots ,{\hat{y}}_{I}^{*}\right)$ is the best solution, and where ${\hat{y}}_{i}^{*}=\left({\hat{l}}_{1}^{*1},\ldots ,{\hat{l}}_{1}^{*j},\ldots ,{\hat{l}}_{1}^{*j}\right)$ is obtained by the CTC decoder, where each ${\hat{l}}_{1}^{*j}$ is the best solution. Based on the cross-entropy loss function [26], the loss function of CTC decoder, $L_{CTC}$ is defined.
(1)$$\begin{array}{c} L_{CTC}\left(Y,\hat{Y}\right)=-\frac{1}{I\times J}\sum_{i=1}^{I}\sum_{j=1}^{J}\;y_{i,j}\times log\left({\hat{y}}_{i,j}\right)=-\frac{1}{I\times J\times K}\sum_{i=1}^{I}\sum_{j=1}^{J}\sum_{k=1}^{K}\;l_{i}^{j}\times log\times \left({\hat{l}}_{j,k}^{i}\right), \end{array}$$
where, $y_{i,j},l_{i}^{j}$ represents the ground-truth label of the *j*-th motion of $x_{i}$, ${\hat{y}}_{i,j}=\left({\hat{l}}_{i,1}^{j},\ldots ,{\hat{l}}_{i,k}^{j},\ldots ,{\hat{l}}_{i,K}^{j}\right)$ is the predicted label sequence for the CTC classification decoder for $x_{i}$ to the *j*-th motion corresponding to *k*-th motion category for all 1≤k≤K, and ${\hat{l}}_{i,k}^{j}$ is the predicted label of the *j*-th motion corresponding to the *k*-th motion category, where 1≤k≤K. Consequently, the best loss function $L_{CTC}^{*}\left(Y,{\hat{Y}}^{*}\right)$ is acquired under the best solution ${\hat{Y}}^{*}$ is obtained.

Next, the Transformer-sequential-decoder loss function $L_{Seq}$ is described below. To make the continuous human motion more effectively learn its temporal correlation, we build a second decoder, called a Transformer-sequential-decoder, in our trained model. The prediction result of the Transformer-sequential-decoder classification is denoted as ${\tilde{Y}}_{i}=\left({\tilde{l}}_{i}^{1},\ldots ,{\tilde{l}}_{i}^{j},\ldots ,{\tilde{l}}_{i}^{J}\right)$, where ${\tilde{l}}_{i}^{j}$ represents the predicted label of the *j*-th motion representation for $x_{i}$, 1≤j≤J, and *J* is a dynamic value for different $x_{i}$, if ${\tilde{y}}_{i}=P\left(y_{i}|x_{i}\right)$ under $P(y_{i}|x_{i})$ is the predicted probability. Therefore, we have a set of predicted label sequences $\tilde{Y}=\left({\tilde{y}}_{1},\ldots ,{\tilde{y}}_{i},\ldots ,{\tilde{y}}_{I}\right)$ for $S=x_{1},\ldots ,x_{i},\ldots ,x_{I}$. The goal is to find the best solution for the classification of the prediction results of the Transformer-sequential decoder ${\tilde{Y}}^{*}=arg\;max\;P\left(Y|S\right)$, where ${\tilde{Y}}^{*}=\left({\tilde{y}}_{1}^{*},\ldots ,{\tilde{y}}_{i}^{*},\ldots ,{\tilde{y}}_{I}^{*}\right)$ is the best classification solution, and ${\tilde{y}}_{i}^{*}=\left({\tilde{l}}_{i}^{*1},\ldots ,{\tilde{l}}_{i}^{*j},\ldots ,{\tilde{l}}_{i}^{*J}\right)$ is obtained using the Transformer-sequential decoder, where each ${\tilde{l}}_{i}^{j}$ is the best solution. Based on the cross-entropy loss function [26], the loss function of the Transformer-sequential decoder, $L_{Seq}$, is defined.
(2)$$\begin{array}{c} L_{Seq}\left(Y,\tilde{Y}\right)=-\frac{1}{I\times J}\sum_{i=1}^{I}\sum_{j=1}^{J}\;y_{i,j}\times log\left({\tilde{y}}_{i,j}\right)=-\frac{1}{I\times J\times K}\sum_{i=1}^{I}\sum_{j=1}^{J}\sum_{k=1}^{K}\;l_{i}^{j}\times log\times \left({\tilde{l}}_{i,k}^{j}\right), \end{array}$$
where, $y_{i,j}$, $l_{i}^{j}$ represent the ground-truth label of the *j*-th motion of $x_{i}$, ${\tilde{y}}_{i,j}=\left({\tilde{l}}_{i,1}^{j},\ldots ,{\tilde{l}}_{i,k}^{j},\ldots ,{\tilde{l}}_{i,K}^{j}\right)$ is the predicted label sequence for the Transformer-sequential decoder for $x_{i}$ to the *j*-th motion corresponding to *k*-th motion category for all 1≤k≤K, and ${\tilde{l}}_{i,k}^{j}$ is the predicted label of the *j*-th motion corresponding to the *k*-th motion category, where 1≤k≤K. Consequently, the best loss function $L_{Seq}^{*}\left(Y,{\tilde{Y}}^{*}\right)$ acquired under the best solution ${\tilde{Y}}^{*}$ is obtained.

The loss $L_{Imp}$ of the Transformer-importance-decoder is described below. To handle the classification of continuous human motions with high similarity, we build a third decoder, called a Transformer-importance-decoder, in our trained model. Assume that the prediction result of the Transformer-importance-decoder classification of $x_{i}$ is ${\bar{y}}_{i}=\left({\bar{l}}_{i}^{1},\ldots ,{\bar{l}}_{i}^{j},\ldots ,{\bar{l}}_{i}^{J}\right)$, where ${\bar{l}}_{i}^{j}$ representing the predicted label of the *j*-th motion representation for $x_{i}$, for 1≤j≤J, ${\bar{y}}_{i}=P\left(y_{i}|x_{i}\right)$ under $P(y_{i}|x_{i})$ is the predicted probability. Therefore, we have a set of predicted label sequences $\bar{Y}=\left({\bar{y}}_{1},\ldots ,{\bar{y}}_{i},\ldots ,{\bar{y}}_{I}\right)$ for $S=x_{1},\ldots ,x_{i},\ldots ,x_{I}$. The goal is to find the best solution for the classification of the prediction results of the Transformer-importance-decoder if ${\bar{Y}}^{*}=arg\;max\;P\left(Y|S\right)$, where ${\bar{Y}}^{*}=\left({\bar{y}}_{1}^{*},\ldots ,{\bar{y}}_{i}^{*},\ldots ,{\bar{y}}_{I}^{*}\right)$ is the best classification solution ${\bar{Y}}^{*}=\left({\bar{l}}_{i}^{*1},\ldots ,{\bar{l}}_{i}^{*j},\ldots ,{\bar{l}}_{i}^{*J}\right)$ obtained using the Transformer-importance-decoder, and ${\bar{l}}_{i}^{*j}$ is the best solution. Based on the cross-entropy loss function [26], the loss function of the Transformer-importance-decoder, $L_{Imp}$ is defined.
(3)$$\begin{array}{c} L_{Imp}\left(Y,\bar{Y}\right)=-\frac{1}{I\times J}\sum_{i=1}^{I}\sum_{j=1}^{J}\;y_{i,j}\times log\left({\bar{y}}_{i,j}\right)=-\frac{1}{I\times J\times K}\sum_{i=1}^{I}\sum_{j=1}^{J}\sum_{k=1}^{K}\;l_{i}^{j}\times log\times \left({\bar{l}}_{i,k}^{j}\right), \end{array}$$
where, $y_{i,j}$, $l_{i}^{j}$ represents the ground-truth label of the *j*-th motion of $x_{i}$, ${\bar{y}}_{i,j}=\left({\bar{l}}_{i,1}^{j},\ldots ,{\bar{l}}_{i,k}^{j},\ldots ,{\bar{l}}_{i,K}^{j}\right)$ is the predicted label sequence for the Transformer-importance-decoder for $x_{i}$ to the *j*-th motion corresponding to *k*-th motion category for all 1≤k≤K, and ${\bar{l}}_{i,k}^{j}$ is the predicted label of the *j*-th motion corresponding to the *k*-th motion category, where 1≤k≤K. Consequently, the best loss function $L_{Imp}^{*}\left(Y,{\bar{Y}}^{*}\right)$ acquired under the best solution ${\bar{Y}}^{*}$ is obtained.

Finally, the overall best loss function $L_{CHMR}^{*}$ is
(4)$$\begin{array}{c} L_{CHMR}^{*}=\alpha \times L_{CTC}^{*}\left(Y,{\hat{Y}}^{*}\right)+\beta \times L_{Seq}^{*}\left(Y,{\tilde{Y}}^{*}\right)+\gamma \times L_{Imp}^{*}\left(Y,{\bar{Y}}^{*}\right), \end{array}$$
where α, β, and γ are the tunable parameters and 0≤α≤1, 0≤β≤1, 0≤γ≤1, and $L_{CTC}^{*}\left(Y,{\hat{Y}}^{*}\right)$ is the best loss function of CTC decoder, $L_{Seq}^{*}\left(Y,{\tilde{Y}}^{*}\right)$ is the best loss function of the Transformer-sequential-decoder, and $L_{Imp}^{*}\left(Y,{\bar{Y}}^{*}\right)$ is the best loss function of the Transformer-importance-decoder.

### 3.3. Basic Idea

One novel CHMR result [11] extracts the motion features through SENe; BLSTM is used as an encoder to learn the time series feature, and the Connectionist Temporal Classification (CTC) [14] is then utilized as a decoder to solve blank and featureless segmentation problems. Figure 4a shows this method. One other novel result for recognizing single gesture motion is from [13], where a Transformer-encoder architecture is used by learning the temporal features from time series of the Range-Doppler Map (RDM), Range Azimuth Map (RAM), and Range Elevation Map (REM), and these temporal features are fused as a fused feature. Figure 4b displays the method. In the paper, the temporal features of the time series of the Range-Doppler Map (RDM), Range Azimuth Map (RAM), and Range Elevation Map (REM) are simplified as 3D features.

In our proposed Transformer-encoder model, the Doppler-based feature, angle-based feature, and range-based feature will be independently learnt as different tasks. In our proposed Transformer-decoder model, the Doppler-based feature, angle-based feature, and range-based features will be sequentially decoded by order through a cross-attention mechanism, called the Transformer-sequential-decoder. A joint decoding is then used to combine the loss value from CTC [14] and two loss values from our proposed Transformer-sequential-decoder model for the recognition optimization. Figure 4c illustrates this concept.

The importance ratio of the Doppler-based feature, angle-based feature, and range-based features for each different time segment of the observed time period should be different, as shown in Figure 5. The importance ratio of the Doppler-based feature, angle-based feature, and range-based features of each time segment will be learned in our design algorithm. This importance ratio is used to significantly improve our CHMR result, especially for the high-similarity motions.

## 4. Our Proposed CHMR-HS Algorithm

In this section, our Transformer-Based Learning with High Similarity for CHMR is divided into four phases, as follows. The flowchart of the proposed the algorithm is also given in Figure 6.
**(1)** *Data pre-processing* phase: The main work is to obtain a wider range of field-of-view data, eliminate the radar signal noise, extract the distance frequency and Doppler frequency information, and reduce the noise frequency generated by the Doppler effects by extracting meaningful 2D and 3D features.**(2)** *Multi-feature embedding* phase: This phase is to embed 2D and 3D features through the CNN embedding module into our designed network.**(3)** *Transformer-based encoding* phase: This phase is to encode 2D and 3D embedded features, including positional embedding, Transformer-encoder, and feature fusion operations, into the Transformer-based network.**(4)** *Transformer-sequential-decoding* phase: This phase describes how to perform CTC-decoder, Transformer-sequential-decoder, and Transformer-importance-decoder, and how to obtain the best sequence-to-sequence result from joint-decoding among three decoders.

### 4.1. Data Pre-processing Phase

The data pre-processing phase is to describe how to obtain *S*, which is a set of training samples of radar images, $S=x_{1},\ldots ,x_{i},\ldots ,x_{I}$ by acquiring FMCW radar data and processing the digital signal converted by a hardware signal processor, as an input of our algorithm, 1≤i≤I, where $x_{i}$ can be a sample of a 2D radar feature or a 3D radar feature. If $x_{i}\in S_{2D}$, each $x_{i}$ contains the *i*-th training sample of the Doppler signature, angle signature, and range signature. If $x_{i}\in S_{3D}$, each $x_{i}$ contains *i*-th training sample of the range-Doppler signature and the range-angle signature.

In the following, we will describe how to obtain $x_{i}\in S_{3D}$ or $S_{2D}$. The continuous *T* FMCW signal samples can be expressed as, $S_{FMCW}^{t}\in R^{m\times n\times T}$, for 1≤t≤T, where $S_{FMCW}^{t}$ is represented as the *t*-th continuous FMCW radar signals, where *m* is the sampling chirp number, *n* is the received antenna number, and *T* is the chirp number times the chirp duration time. If $x_{i}\in S_{3D}$, $x_{i}$ consists of multi-features, including a *T* number of range-Doppler signatures, denoted as $RDS_{T}$, *T* number of range-Doppler signatures, denoted as $RAS_{T}$, and *T* number of range-elevation signatures, denoted as $RES_{T}$, which are extracted from continuous *T* FMCW signal samples $S_{FMCW}^{t}$, for 1≤t≤T, we let $x_{i}=RDS_{T},RAS_{T},RES_{T}\in S_{3D}$. If $x_{i}\in S_{2D}$, $x_{i}$ consists of multi-features, including a *T* number of Doppler signatures, denoted as $DS_{T}$, *T* number of range signatures, denoted as $RS_{T}$, *T* number of azimuth signatures, denoted as $AS_{T}$, and *T* number of elevation signatures, denoted as $ES_{T}$, which are also extracted from the continuous *T* FMCW signal samples $S_{FMCW}^{t}$, for 1≤t≤T, we let $x_{i}$ = $\left\{DS_{T},RS_{T},AS_{T},ES_{T}\right\}\in S_{2D}$. Consequently, the output of the data pre-processing phase is to extract $x_{i}=\left\{RDS_{T},RAS_{T},RES_{T}\right\}\in S_{3D}$ or $x_{i}=\left\{DS_{T},RS_{T},AS_{T},ES_{T}\right\}\in S_{2D}$ from the continuous *T* FMCW signal samples, $S_{FMCW}^{t}$, for 1≤t≤T, as follows.

To obtain a wider range of field-of-view data, a Binary Phase Modulation (BPM) [27], or called as BPM-MIMO scheme, is firstly executed by utilizing the FMCW MIMO, by applying the BPM function [27] to yield $S_{FMCW}^{t}$, as $S_{m,n,c}^{t}=BPM\left(S_{FMC}^{t}\right)$, for 1≤t≤T, under $S_{FMCW}^{t}$, for 1≤t≤T, is the continuous *T* FMCW signal samples, where $S_{m,n,c}^{t}\in R^{m\times n\times c}$, where 1≤t≤T, each is represented as the m×n×c 3-dimensional data matrix, where *m* is the sampling chirp number, *n* is the received antenna number, and *c* is the chirp number utilized after performing the BPM algorithm.

To eliminate the radar signal noise, a Moving Target Indication (MTI) filtering algorithm [16] is then utilized, the MTI noise filtering function is performed on $S_{m,n,c}^{t}$ to obtain $M_{m,n,c}^{t}$, where 1≤t≤T, such that $M_{m,n,c}^{t}=MTI\left(S_{m,n,c}^{t}\right)$, where $M_{m,n,c}^{t}$ is represented as an m×n×c three-dimensional data matrix after performing the MTI noise filtering function to eliminate the radar signal noise, where 1≤t≤T. To avoid the different background issue, we adopt the MTI (Moving Target Indication) filter algorithm in our paper. The signal of the non-moving object (no phase difference) will be removed to ensure that the detected object is only detected in a moving condition, such that the movement, or called as phase difference, can be directly retained. We may also call this the static removal operation.

To extract the distance frequency and the Doppler frequency information of the detected objects, a Fast Fourier Transform (FFT) algorithm [4] is applied to the $M_{m,n,c}^{T}$ data matrix to obtain ${\dot{F}}_{m,n,c}^{T}$ data matrix, such that ${\dot{F}}_{m,n,c}^{t}=FFT\left(M_{m,n,c}^{t}\right)$, where 1≤t≤T, by performing the FFT procedure on each column of $M_{m,n,c}^{t}$, where 1≤t≤T. Then, a Fast Fourier Transform (FFT) [4] algorithm is performed on ${\dot{F}}_{m,n,c}^{t}$ to obtain the ${\ddot{F}}_{m,n,c}^{t}$ data matrix, such that ${\ddot{F}}_{m,n,c}^{t}=FFT\left({\dot{F}}_{m,n,c}^{t}\right)$, where 1≤t≤T, by performing the FFT procedure on each row of ${\dot{F}}_{m,n,c}^{t}$, where 1≤t≤T.

To reduce the noise frequency generated by the Doppler effects, a Constant False Alarm Rate (CFAR) filtering algorithm [17] is also utilized for the FFT data matrix ${\ddot{F}}_{m,n,c}^{T}$ to acquire a new data matrix $C_{m,n,c}^{T}$, such that $C_{m,n,c}^{t}=CFAR\left({\ddot{F}}_{m,n,c}^{t}\right)$, where 1≤t≤T.

The purpose of the background denoising algorithm is to extract the phase frequency between the signals. In our work, we only take the data frames with the largest signals after the Fourier transformation, and then concatenate these data frames together. The intuition is a continuous echo map of the torso. If one or two noises are larger than the torso reflections, or there is a small noise in the process of denoising, it does not matter for the misguided training, since one or two noises of the tens to hundreds of data frames will be moderately retained and will not be particularly problematic.

In the following, we describe how to obtain $x_{i}=\left\{RDS_{T},RAS_{T},RES_{T}\right\}\in S_{3D}$ from the $C_{m,n,c}^{t}$, where 1≤t≤T. We show how to generate $RDS_{T}=\left\{RDM_{m,c}^{1},\ldots ,RDM_{m,c}^{t},\ldots ,RDM_{m,c}^{T}\right\}$, 1≤t≤T, from $C_{m,n,c}^{T}$, as follows. We let $RDM_{m,c}^{t}$ be an m×n two-dimensional data matrix using $RDM_{m,c}^{t}=\sum_{k=1}^{n}C_{m,k,c}^{t}$, so the range-Doppler signature $RDS_{T}=\left\{RDM_{m,c}^{1},\ldots ,RDM_{m,c}^{t},\ldots ,RDM_{m,c}^{T}\right\}$, where 1≤t≤T, is finally obtained. Based on the beamforming [28] processing principle, we may obtain the range-azimuth signature, $RAS_{T}=\left\{RAM_{m,c}^{1},\ldots ,RAM_{m,c}^{t},\ldots ,RAM_{m,c}^{T}\right\}$, where 1≤t≤T, and range-elevation signature, $RES_{T}=\left\{REM_{m,c}^{1},\ldots ,REM_{m,c}^{t},\ldots ,REM_{m,c}^{T}\right\}$, where 1≤t≤T. A beamforming function *B*, Reference [28] is applied to extract the angle frequency information from the received antennas; $BA_{m,c}^{t}$ and $BE_{m,c}^{t}$ are obtained, where $BA_{m,c}^{t}=B\left({\dot{F}}_{m,1,c}^{t},{\dot{F}}_{m,2,c}^{t},\ldots ,{\dot{F}}_{m,j,c}^{t}\right)$ and $BE_{m,c}^{t}=B\left({\dot{F}}_{m,1,c}^{t},{\dot{F}}_{m,2,c}^{t},\ldots ,{\dot{F}}_{m,j,c}^{t}\right)$, where the first to *j*-th azimuth antenna data are ${\dot{F}}_{m,1,c}^{t},{\dot{F}}_{m,2,c}^{t},...,{\dot{F}}_{m,j,c}^{t}$ of ${\dot{F}}_{m,n,c}^{t}$, and (j+1)-th to *n*-th elevation antenna data are ${\dot{F}}_{m,j+1,c}^{t},{\dot{F}}_{m,j+2,c}^{t},...,{\dot{F}}_{m,n,c}^{t}$ of ${\dot{F}}_{m,n,c}^{t}$, where 1≤t≤T. Finally, $RAM_{m,c}^{t}=CFAR\left(BA_{m,c}^{t}\right)$ and $REM_{m,c}^{t}=CFAR\left(BE_{m,c}^{t}\right)$ are obtained, where 1≤t≤T.

Figure 7 shows an example for our range-Doppler signature $RDS_{T}$, range-azimuth signature $RAS_{T}$, and range-elevation signature $RES_{T}$.

In the following, we describe how to obtain $x_{i}$ = $\left\{DS_{T},RS_{T},AS_{T},ES_{T}\right\}\in S_{2D}$ from $\left\{RDS_{T},RAS_{T},RES_{T}\right\}\in S_{3D}$. First, we show how to obtain the Doppler signature $DS_{T}=\left\{D_{m}^{1},\ldots ,D_{m}^{t},\ldots ,D_{m}^{T}\right\}$,where 1≤t≤T. Given ${\dot{F}}_{m,n,c}^{t}=FFT\left(M_{m,n,c}^{t}\right)$, where 1≤t≤T, we let $D_{m}^{t}=\sum_{i=1}^{n}STFT\left(\sum_{j=1}^{c}{\dot{F}}_{m,i,j}^{t}\right)$, by performing the Short-Time Fourier Transform (STFT) operation on $\sum_{i=1}^{n}STFT\left(\sum_{j=1}^{c}{\dot{F}}_{m,i,j}^{t}\right)$, where 1≤t≤T. Next, the range signature $RS_{T}=\left\{R_{m}^{1},\ldots ,R_{m}^{t},\ldots ,R_{m}^{T}\right\}$ is constructed using $R_{m}^{t}=\sum_{i=1}^{c}RDM_{m,i}^{t}$, where $R_{m}^{t}$ is the one-dimensional range data of *m* sampling chirp numbers within *t* sampling time, under $RDS_{T}=\left\{RDM_{m,c}^{1},\ldots ,RDM_{m,c}^{t},\ldots ,RDM_{m,c}^{T}\right\}$, where 1≤t≤T. The azimuth signature $AS_{T}=\left\{A_{m}^{1},\ldots ,A_{m}^{t},\ldots ,A_{m}^{T}\right\}$ is then constructed, such that $A_{m}^{t}=\sum_{i=1}^{c}RAM_{m,i}^{t}$, where $A_{m}^{t}$ is the one-dimensional azimuth data of *m* sampling chirp numbers within *t* sampling time, under a range-azimuth signature $RAS_{T}=\left\{RAM_{m,c}^{1},\ldots ,RAM_{m,c}^{t},\ldots ,RAM_{m,c}^{T}\right\}$, where 1≤t≤T. Finally, the elevation signature $ES_{T}=\left\{E_{m}^{1},\ldots ,E_{m}^{t},\ldots ,E_{m}^{T}\right\}$ is formed by $E_{m}^{t}=\sum_{i=1}^{c}REM_{m,i}^{t}$, where $E_{m}^{t}$ is the one-dimensional elevation data of *m* sampling chirp numbers within *t* sampling time, under the range-elevation signature $RES_{T}=\left\{REM_{m,c}^{1},\ldots ,REM_{m,c}^{t},\ldots ,REM_{m,c}^{T}\right\}$, where 1≤t≤T.

An example is shown in Figure 8, with the generation of 2D feature signatures of the Doppler signature $DS_{T}$, range signature $RS_{T}$, azimuth signature $AS_{T}$, and elevation signature $ES_{T}$. An example of CHM is given in Figure 9a for the human motions of wave, jumping forward, and the vertical-motion group (including falling and lying down). Figure 9b provides the human motions of the walking group (including walking forward, walking backward, walking left, and walking right). Figure 9c displays the human motions of the running group (including running forward, running backward, running left, and running right). Figure 10a give an examples of continuous human motions with 2D feature signatures under four motions in 12 s, and Figure 10b shows another example of two motions in 12 s.

### 4.2. Multi-Feature Embedding Phase

The task of the multi-feature embedding phase is to embed 2D features $x_{i}=\left\{DS_{T},RS_{T},AS_{T},ES_{T}\right\}$$\in S_{2D}$, and $S=\{x_{1},...,x_{i},...,x_{I}\}$, where $DS_{T}=\left\{D_{m}^{1},\ldots ,D_{m}^{t},\ldots ,D_{m}^{T}\right\}$, $RS_{T}=\left\{R_{m}^{1},\ldots ,R_{m}^{t},\ldots ,R_{m}^{T}\right\}$, $AS_{T}=\left\{A_{m}^{1},\ldots ,A_{m}^{t},\ldots ,A_{m}^{T}\right\}$, and $ES_{T}=\left\{E_{m}^{1},\ldots ,E_{m}^{t},\ldots ,E_{m}^{T}\right\}$, where 1≤t≤T, through a CNN embedding module into our designed network.

In the following, we describe how to perform the 2D feature data embedding. Let a CNN embedding module be represented as $E_{CNN}^{2D}$, and each 2D feature from $\left\{DS_{T},RS_{T},AS_{T},ES_{T}\right\}$ is independently embedded by a different $E_{CNN}^{2D}$. In this work, $E_{CNN}^{2D}$ consists of a first convolution layer of 3×1×16, with stride of 3×1, batch norm., ReLU, a max pooling layer with a kernel size of 3×1, batch norm., ReLU, a second convolution layer by 3×1×16 with stride of 3×1, batch norm., ReLU, max pooling layer with a kernel size of 5×1, batch norm., ReLU, a third convolution layer by 3×1×16 with stride of 3×1, batch norm., ReLU, and an average pooling is finally used to extract all features. Note that the 2D feature data embedding $E_{CNN}^{2D}$ aim to embed the spatial information from 2D features $x_{i}=\left\{DS_{T},RS_{T},AS_{T},ES_{T}\right\}$. an example of $E_{CNN}^{2D}$ can be seen in Figure 11.

The 2D feature data embedding operations performed through the CNN embedding module $E_{CNN}^{2D}$ are given as follows.
(5)$$\left\{\begin{array}{c} DS_{E}=E_{CNN}^{2D}\left(DS_{T}\right),\\ RS_{E}=E_{CNN}^{2D}\left(RS_{T}\right),\\ AS_{E}=E_{CNN}^{2D}\left(AS_{T}\right),\\ ES_{E}=E_{CNN}^{2D}\left(ES_{T}\right), \end{array}\right.$$
where $DS_{E},RS_{E},AS_{E},ES_{E}$ are the embedded features for the 2D original features $DS_{T},RS_{T},AS_{T},ES_{T}$, where the data size of each embedded feature is m×T.

Figure 11a gives an example of $DS_{E},RS_{E},AS_{E}$, and $ES_{E}$, which are embedded features for the 2D features $DS_{T},RS_{T},AS_{T}$, and $ES_{T}$.

Another task of the multi-feature embedding phase is to embed the 3D features $x_{i}=\left\{RDS_{T},RAS_{T},RES_{T}\right\}\in S_{3D}$, and $S=\{x_{1},...,x_{i},...,x_{I}\}$, where $RDS_{T}=\left\{RDM_{m,c}^{1},\ldots ,RDM_{m,c}^{t},\ldots ,RDM_{m,c}^{T}\right\}$, $RAS_{T}=\left\{RAM_{m,c}^{1},\ldots ,RAM_{m,c}^{t},\ldots ,RAM_{m,c}^{T}\right\}$, and $RES_{T}=\left\{REM_{m,c}^{1},\ldots ,REM_{m,c}^{t},\ldots ,REM_{m,c}^{T}\right\}$, where 1≤t≤T, through the CNN network into our designed network. In the following, we describe how to perform the 3D feature data embedding. Let a CNN embedding module be represented as $E_{CNN}^{3D}$, and each 3D feature from $\left\{RDS_{T},RAS_{T},RES_{T}\right\}$ is independently embedded by a different $E_{CNN}^{3D}$. In this work, $E_{CNN}^{3D}$ consists of a first convolution layer by 3×3×16 with stride of 1×1, batch norm., ReLU, max pooling layer with kernel size of 2×2 with stride of 1×1, batch norm., ReLU, a second convolution layer by 3×3×16 with a stride of 1×1, batch norm., ReLU, max pooling layer with kernel size of 2×2 with stride of 2×2; and then an average pooling is used to extract all the features. Finally, a Multilayer Perceptron (MLP) is additionally used to embed the features into the temporal space. Note that the purpose of the 3D feature data embedding $E_{CNN}^{2D}$ aims to first embed the 3D features into the spatial space, and MLP is then used to embed the 3D features into the spatial space. An example of $E_{CNN}^{3D}$ can be seen in Figure 12.

The 3D feature data embedding operations performed through the CNN embedding module $E_{CNN}^{3D}$ is given as follows:(6)$$\left\{\begin{array}{c} RDS_{E}=E_{CNN}^{3D}\left(RDS_{T}\right),\\ RAS_{E}=E_{CNN}^{3D}\left(RAS_{T}\right),\\ RES_{E}=E_{CNN}^{3D}\left(RES_{T}\right), \end{array}\right.$$
where $RDS_{E},RAS_{E}$, and $RES_{E}$ are the embedded features for the 3D original features $RDS_{T},RAS_{T}$, and $RES_{T}$, where the data size of each embedded feature is m×c×T.

Figure 12a gives an example of $RDS_{E},RAS_{E}$, and $RES_{E}$, embedded features for the 3D features $RDS_{T},RAS_{T}$, and $RES_{T}$.

### 4.3. Transformer-Based Encoding Phase

In the multi-feature embedding phase, the 2D embedded features $DS_{E},RS_{E},AS_{E},ES_{E}$, or the 3D embedded features $RDS_{E},RAS_{E}$, and $RES_{E}$ are embedded into our Transformer-based network, then we describe how to perform our Transformer-based encoding procedure. This phase is divided into the positional embedding operation, Transformer-encoder operation, and feature fusion operation, which are described as follows. It is observed that we let each different 2D embedded feature or different 3D embedded feature be embedded into the disjoint Transformer encoder, and all of the different embedding features among the different disjoint Transformer encoders will be merged together in the feature fusion operation.

In the following, we describe the positional embedding operation, based on our 2D/3D embedded features. Let $T_{pos}$ be the position embedding function [13] for the 2D embedded features $DS_{E},RS_{E},AS_{E},$ and $ES_{E}$.
(7)$$\left\{\begin{array}{c} DS_{P}=T_{pos}\left(DS_{E}\right)\\ RS_{P}=T_{pos}\left(RS_{E}\right)\\ AS_{P}=T_{pos}\left(AS_{E}\right)\\ ES_{P}=T_{pos}\left(ES_{E}\right), \end{array}\right.$$
where $DS_{p},RS_{p},AS_{p},$ and $ES_{p}$ are position embedding after performing the $T_{pos}$ function to the 2D embedded features $DS_{E},RS_{E},AS_{E}$, and $ES_{E}$, respectively. We denote the 2D position embedding feature as $F_{pos}^{2D}\in \left\{DS_{P},RS_{P},AS_{P},ES_{P}\right\}$.

Let $T_{pos}$ be the position embedding function for the 3D embedded features $RDS_{E}$, $RAS_{E}$, and $RES_{E}$, as follows.
(8)$$\left\{\begin{array}{c} RDS_{P}=T_{pos}\left(RDS_{E}\right)\\ RAS_{P}=T_{pos}\left(RAS_{E}\right)\\ RES_{P}=T_{pos}\left(RES_{E}\right), \end{array}\right.$$
where $RDS_{E}$, $RAS_{E}$, and $RES_{E}$ are position embedded after performing the $T_{pos}$ function to the 3D embedded features $RDS_{E},RAS_{E}$, and $RES_{E}$, respectively. We denote the 3D position embedding feature, $F_{pos}^{3D}\in \left\{RDS_{P},RAS_{P},RES_{P}\right\}$. The $T_{pos}$ function follows the existing position embedding function from [13], such that $T_{pos}=\left\{\begin{array}{c} PE_{(}pos,2i)=\sin\left(\mathrm{pos}/{10,000}^{2i/d_{\mathrm{model}}}\right)\\ PE_{(}pos,2i+1)=\cos\left(\mathrm{pos}/{10,000}^{2i/d_{\mathrm{model}}}\right). \end{array}\right.$

Next, we describe the Transformer-encoder operation by applying our 2D/3D embedded features. Let the 2D position embedding feature be $F_{pos}^{2D}\in \left\{DS_{P},RS_{P},AS_{P},ES_{P}\right\}$, and the 3D position embedding feature be $F_{pos}^{3D}\in \left\{RDS_{P},RAS_{P},RES_{P}\right\}$. We then utilize a Transformer encoder, $T_{Enc}$, for the 2D position embedding feature, $F_{pos}^{2D}\in \left\{DS_{P},RS_{P},AS_{P},ES_{P}\right\}$.
(9)$$\begin{array}{c} F_{Enc}^{2D}=T_{Enc}\left(F_{pos}^{2D}\right), \end{array}$$
where $F_{Enc}^{2D}\in \left\{RS_{Enc},RS_{Enc},AS_{Enc},RS_{Enc}\right\}$ is the 2D embedding feature after performing the Transformer encoder for $F_{pos}^{2D}\in \left\{RS_{P},RS_{P},AS_{P},RS_{P}\right\}$. Similarly, we then also utilize a Transformer encoder, $T_{Enc}$, for the 3D position embedding feature, $F_{pos}^{3D}\in \left\{RDS_{P},RAS_{P},RES_{P}\right\}$, as follows.
(10)$$\begin{array}{c} F_{Enc}^{3D}=T_{Enc}\left(F_{pos}^{3D}\right), \end{array}$$
where $F_{Enc}^{3D}\in \left\{RDS_{Enc},RAS_{Enc},RES_{Enc}\right\}$ is the 3D embedding feature after performing the Transformer encoder for $F_{pos}^{3D}\in \left\{RDS_{P},RAS_{P},RES_{P}\right\}$. The Transformer encoder $T_{Enc}$ follows the existing result from [13], which is to calculate the linear transformation of the self-attention weight matrices, the key and value of $F_{pos}^{2D}$ (or $F_{pos}^{3D}$). Let *Q*, *K*, and *V* be the calculation matrices of $F_{pos}^{2D}$ (or $F_{pos}^{3D}$), and let $W^{Q}$, $W^{K}$, and $W^{V}$ be the linear transformation matrices, while the self-attention operation matrices with $QW^{Q}$, $KW^{K}$, and $VW^{V}$ corresponding to $F_{pos}^{2D}$ (or $F_{pos}^{3D}$), are obtained through the linear mapping, normalized by the softmax function. The multi-head features $head_{i}$ are obtained using multi-head self-attention features; the self-attention-based encoder, denoted as $T_{Enc}$, from [13], is expressed as follows.
(11)$$\begin{array}{c} T_{Enc}=\left\{\begin{array}{c} MHA\left(Q,K,V\right)=\left(head_{1}⊕head_{2}⊕\cdots ⊕head_{i}\right)W^{O}\\ head_{i}=SA_{i}\left(QW_{i}^{Q},KW_{i}^{K},VW_{i}^{V}\right),i=1,2,\ldots ,h\\ SA_{i}\left(QW_{i}^{Q},KW_{i}^{K},VW_{i}^{V}\right)=\\ \;\;\;\;\;\;\;softmax(\left(\frac{\left(QW_{i}^{Q}\right){\left(KW_{i}^{K}\right)}^{T}}{\sqrt{d_{k}}}\right)VW_{i}^{V}),i=1,2,\ldots ,h \end{array}\right. \end{array}$$
where $SA_{i}\left(QW_{i}^{Q},KW_{i}^{K},VW_{i}^{V}\right)$ is the feature signature from the linear transformation matrices $W_{i}^{Q}$,$W_{i}^{K}$, and $W_{i}^{V}$,where 1≤i≤h and *h* is the weight between the features, after the dot product operation of the number of layers *Q* and *K* in the multi-head operation, and is divided by the variance value $\sqrt{d_{k}}$ to obtain the weight matrix between 0 and 1; the weight is extracted through the normalization of the softmax function. Finally, the *i*-th self-attention $head_{i}$ is obtained through multiplication by *V*, and the multiple self-attention $head_{1},\ldots ,head_{i},\ldots ,head_{h}$ are concatenated to produce the final self-attention output features.

In the following, we describe how to perform the feature fusion operation. After the Transformer-encoder operation, 2D position embedding feature, $F_{pos}^{2D}\in \left\{DS_{P},RS_{P},AS_{P},ES_{P}\right\}$ and the 3D position embedding feature $F_{pos}^{3D}\in \left\{RDS_{P},RAS_{P},RES_{P}\right\}$ are obtained from different Transformer encoders. All of these disjoint embedded features should be merged together before our proposed Transformer-based decoder. Let $MF^{2D}$ be denoted as the merged 2D embedded features for all 2D position embedding features; we have $MF^{2D}=DS_{Enc}⊕RS_{Enc}⊕AS_{Enc}⊕ES_{Enc}$. Let $MF^{3D}$ denoted as the merged 3D embedded features for all 3D position embedding feature, we have $MF^{3D}=RDS_{Enc}⊕RAS_{Enc}⊕RES_{Enc}$. Finally, we further utilize the multi-layer perceptron MLP, and the average pooling layer AvgPool, for the feature training and feature extraction, as follows. $FF^{2D}=MLP\left(Avgpool\left(MF^{2D}\right)\right)$, $FF^{3D}=MLP\left(Avgpool\left(MF^{2D}\right)\right)$. Note that $FF^{2D}$ or $FF^{3D}$ are simplified as FF for the following explanations, where $FF^{2D}\in R^{m\times c}$ and $FF^{3D}\in R^{m\times c}$.

Examples of the position embedding of the 2D signature, $DS_{P}$, and the 3D signature, $RDS_{P}$, are given in Figure 11b and Figure 12b. The Transformer-encoder operations are performed to have $DS_{Enc}$ and $RDS_{Enc}$, as shown in Figure 11b and Figure 12b. The fusion features $FF^{2D}$ and $FF^{3D}$ are obtained in Figure 11c and Figure 12c.

### 4.4. Transformer-Sequential-Decoding Phase

For the fused feature FF, where $FF\in FF^{2D}$ or $FF^{3D}$ after performing the Transformer-based encoder phase, the proposed Transformer-sequential-decoding phase is then performed. The proposed Transfer-sequential-decoding phase consists of the CTC-decoder, Transformer-sequential-decoder, and Transformer-importance-decoder; each decoder produces a set of predicted label sequences, where $\hat{Y}=\left({\hat{y}}_{1},\ldots ,{\hat{y}}_{i},\ldots ,{\hat{y}}_{I}\right)$, $\tilde{Y}=\left({\tilde{y}}_{1},\ldots ,{\tilde{y}}_{i},\ldots ,{\tilde{y}}_{I}\right)$, and $\bar{Y}=\left({\bar{y}}_{1},\ldots ,{\bar{y}}_{i},\ldots ,{\bar{y}}_{I}\right)$ are sets of predicted label sequences of CTC-decoder, Transformer-sequential-decoder, and Transformer-importance-decoder, under the ground-true label sequence $Y=(y_{1},\ldots ,y_{i},\ldots ,y_{I})$ for all $y_{i}=\left(l_{i}^{1},\ldots ,l_{i}^{j},\ldots ,l_{i}^{J}\right)$ for $x_{i}$ and $S=\{x_{1},\ldots ,x_{i},\ldots ,x_{I}\}$. We will describe how to generate these predicted label sequences from the CTC-decoder, Transformer-sequential-decoder, and Transformer-importance-decoder. Finally, a joint-decoding operation is proposed to jointly combine these predicted label sequences to provide a best predicted label sequence, which is nearer to an optimal solution. First, the CTC decoder is described.

**CTC decoder:** The fused feature FF, where $FF\in FF^{2D}$ or $FF^{3D}$, occurs after performing the Transformer-based encoder phase; the proposed Transfer-sequential-decoding phase is then performed. As mentioned in Section 3.2, the connectionist temporal classification (CTC) [14] technique is utilized to overcome the blank segment and meaningless feature sequence for the long-time series motion recognition; the CTC technique is integrated into our scheme as the first one feature decoder. Let a set of predicted label sequences $\hat{Y}=\left({\hat{y}}_{1},\ldots ,{\hat{y}}_{i},\ldots ,{\hat{y}}_{I}\right)$, for all ${\hat{y}}_{i}=\left({\hat{l}}_{i}^{1},\ldots ,{\hat{l}}_{i}^{j},\ldots ,{\hat{l}}_{i}^{f}\right)=CTC\left(FF\right)$ for $x_{i}$, where the fused feature of $x_{i}$ is $FF\in FF^{2D}$ or $FF^{3D}$. As shown in Figure 13c, the task of the CTC decoder is performed using the fusion feature FF to obtain $\hat{Y}=\left({\hat{y}}_{1},\ldots ,{\hat{y}}_{i},\ldots ,{\hat{y}}_{I}\right)$, and the loss $L_{CTC}$ can be calculated via Equation (Equation 1) under the predicted label sequence $\hat{Y}$ and the ground-true label sequence *Y*.**Transformer-sequential-decoder:** To allow the continuous human motion to more effectively learn its temporal correlation, a second decoder, the Transformer-sequential-decoder, is built in our trained model. The output of the Transformer-sequential-decoder is $\tilde{Y}=\left({\tilde{y}}_{1},\ldots ,{\tilde{y}}_{i},\ldots ,{\tilde{y}}_{I}\right)$ for $x_{i}$ and $S=x_{1},\ldots ,x_{i},\ldots ,x_{I}$, where the fused feature of $x_{i}$ is simplified as FF. The basic idea of the Transformer-sequential-decoder is to sequentially decode the Doppler, angle, and range embedded features, and these embedded features are extracted from Transformer-encoder.

The sequential operation of Transformer-sequential-decoder is represented by
(12)$$\begin{array}{c} L_{Dec}^{3}=T_{Dec}^{E}(L_{Enc}^{3},\left(T_{Enc}^{E}\left(L_{Enc}^{2},T_{Dec}\left(L_{Enc}^{1},M\right),E\right),E\right), \end{array}$$
where $L_{Enc}^{1}\in \left\{DS_{Enc}\right\}$, $L_{Enc}^{2}\in \left\{AS_{Enc},ES_{Enc}\right\}$, and $L_{Enc}^{3}\in \left\{RS_{Enc}\right\}$ if 2D features are used; $L_{Enc}^{1}\in \left\{RDS_{Enc}\right\}$ and $L_{Enc}^{2}\in \left\{RAS_{Enc},RES_{Enc}\right\}$, if 3D features are utilized. The additional weight, *E*, is the Doppler feature enhancement matrix, which is obtained by $E=L_{Enc}^{1}\times W^{E}$, where $W^{E}$ is the transformation weight matrix [22] and *M* is the label feature matrix, which is obtained by $M=MHA(Y+Y^{M})$, by performing the mask self-attention operation on the input label *Y* and mask matrix $Y^{M}$ [22] to prevent the mechanism of observing a subsequent time scale problem.

The sequential operation is stated below. The first function $T_{Dec}\left(L_{Enc}^{1},M\right)$ is initially performed, where $T_{Dec}$ is same as $T_{Enc}$ in Equation (Equation 11), if $L_{Enc}^{1}\in DS_{Enc}$ or $RDS_{Enc}$. The output of $L_{Dec}^{1}=T_{Dec}\left(L_{Enc}^{1},M\right)$ is used again for the second function $T_{Dec}^{E}(L_{Enc}^{2}$, $T_{Dec}\left(L_{Enc}^{1},M\right),E)$ if $L_{Enc}^{2}\in \left\{AS_{Enc},ES_{Enc}\right\}$ or $\left\{RAS_{Enc},RES_{Enc}\right\}$ with the additional weight, *E*. Finally, the output of $L_{Dec}^{2}=T_{Dec}^{E}\left(L_{Enc}^{2},T_{Dec}\left(L_{Enc}^{1},M\right),E\right)$ is used again for the third function $L_{Dec}^{3}=T_{Dec}^{E}(L_{Enc}^{3},\left(T_{Enc}^{E}\left(L_{Enc}^{2},T_{Dec}\left(L_{Enc}^{1},M\right),E\right),E\right)$, where $L_{Dec}^{1},L_{Dec}^{2}$, and $L_{Dec}^{3}\in R^{m\times c}$. We finally apply the Multilayer Perceptron (MLP) to decode $\tilde{Y}=MLP\left(L_{Dec}^{3}\right)$, where $\tilde{Y}=\left({\tilde{y}}_{1},\ldots ,{\tilde{y}}_{i},\ldots ,{\tilde{y}}_{I}\right)$. The loss function $L_{Seq}$ is calculated by Equation (Equation 2). The detailed procedures of the Transformer-sequential-decoder module are proposed in Algorithm 1.

Our multi-head attention function, $T_{Dec}^{E}$, modified from [14], is as follows.
(13)$$\begin{array}{c} T_{Dec}^{E}=\left\{\begin{array}{c} EMHA\left(Q,K,V,E\right)=\left(head_{1}⊕head_{2}⊕\cdots ⊕head_{i}\right)W^{O}\\ head_{i}=SA_{i}\left(QW_{i}^{Q},KW_{i}^{K},VW_{i}^{V}\right),i=1,2,...,h\\ SA_{i}\left(QW_{i}^{Q},KW_{i}^{K},VW_{i}^{V}\right)=\\ \;\;\;\;\;\;\;softmax(\left(\frac{{\left(\left({\mathrm{QW}}_{i}^{Q}\right)\left({\mathrm{EW}}_{i}^{E}\right)\right)}^{T}{\left(\left({\mathrm{EW}}_{i}^{E}\right)\left({\mathrm{KW}}_{i}^{K}\right)\right)}^{T}}{\sqrt{d_{k}}}\right)VW_{i}^{V}),i=1,2,\ldots ,h \end{array}\right. \end{array}$$

Note that the key difference is that the weight calculation for self-attention is modified as ${\mathrm{QW}}_{i}^{Q}{\left({\mathrm{EW}}_{i}^{E}\right)}^{T})\left({\mathrm{EW}}_{i}^{E}\right){\left({\mathrm{KW}}_{i}^{K}\right)}^{T}$, where $W_{i}^{E}$ is the weight parameter set for the linear mapping, and *E* is the output feature of the decoder.

Our CHMR-HS architecture with the Transformer-sequential-decoder module, Transformer-importance-decoder module, CTC decoder, and Joint decoding module are illustrated in Figure 13a–d.

Figure 14 displays the detailed operation of the Transformer-sequential-decoder for sequentially decoding the Doppler, Doppler-angle, and Doppler-angle-range features.
**Algorithm 1:** Transformer-sequential-decoder processing.
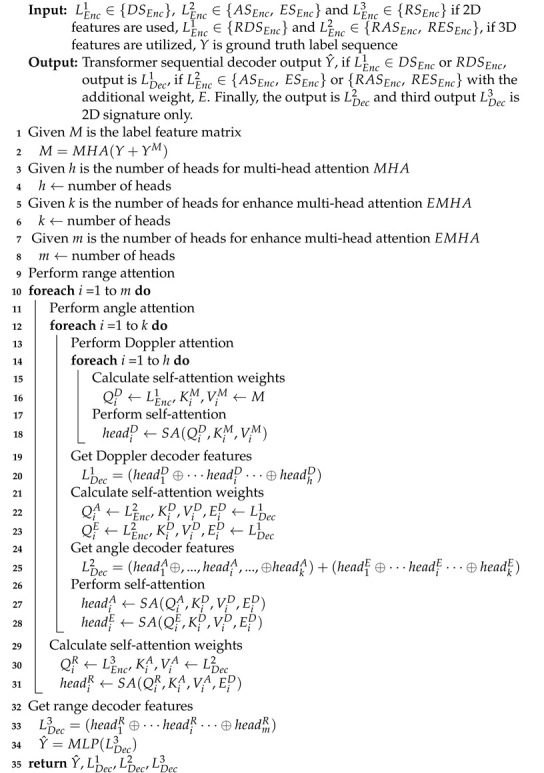


**Transformer-importance-decoder:** In the Transformer-sequential-decoder, the first output $L_{Dec}^{1}=T_{Dec}\left(L_{Enc}^{1},M\right)$ is to only decode the Doppler feature; the second output $L_{Dec}^{2}=T_{Dec}^{E}(L_{Enc}^{2}\left(T_{Dec}\left(L_{Enc}^{1},M\right),E)\right)$ is to sequentially decode the Doppler and angle features. Finally, the third output $L_{Dec}^{3}=T_{Dec}^{E}(L_{Enc}^{3},\left(T_{Enc}^{E}\left(L_{Enc}^{2},T_{Dec}\left(L_{Enc}^{1},M\right),E\right),E)\right)$ is to sequentially decode the Doppler, angle and range features, where $L_{Dec}^{1},L_{Dec}^{2}$, and $L_{Dec}^{3}\in R^{m\times c}$. In the following, we further design a third decoder, called the Transformer-sequential-decoder, to learn the selection from $L_{Dec}^{1}$, or $L_{Dec}^{2}$, or $L_{Dec}^{3}$ among the fixed number of segments, where the fixed number of segment is $\left\lceil \frac{c}{m}\right\rceil$. Note that the fused feature $FF\in R^{m\times c}$, is divided into $\left\lceil \frac{c}{m}\right\rceil$ segments; each segment is an m×m matrix, and $\in R^{m\times m}$. Each segment of these $\left\lceil \frac{c}{m}\right\rceil$ segments should learn its importance to smartly select $L_{Dec}^{1},L_{Dec}^{2},$ or $L_{Dec}^{3}$ as one of inputs to perform the MHA function, where the multi-head self-attention (MHA) function is simplified as the MHA function, based on the expected value of distribution of the Doppler, angle, and range features.

First, we let $FF_{Dec}=MHA\left(FF,M\right)$ to perform the multi-head self-attention (MHA) function to the fused feature FF and the label feature matrix *M*, where $FF\in FF^{2}D$ or $FF^{3D}$, *M* is the label feature matrix, and $FF_{Dec}\in R^{m\times c}$. To estimate the expected value of the distribution of Doppler, angle, and range features, we also calculate $D_{m,c}=MHA\left(L_{Dec}^{1},FF_{Dec}\right)$, $DA_{m,c}=MHA\left(L_{Dec}^{2},FF_{Dec}\right)$, and $DAR_{m,c}=MHA\left(L_{Dec}^{3},FF_{Dec}\right)$, where $DA_{m,c}$, $DAR_{m,c}$, $DAR_{m,c}$ are performed MHA functions to $L_{Dec}^{1},L_{Dec}^{2},$ and $L_{Dec}^{3}$ with $FF_{Dec}$, respectively. We also have the results of $D_{m,c},DA_{m,c},DAR_{m,c}\in R^{m\times c}$, because $FF_{Dec},L_{Dec}^{1},L_{Dec}^{2}$, and $L_{Dec}^{3}\in R^{m\times c}$.

If $D_{m,c},DA_{m,c},DAR_{m,c}\in R^{m\times c}$, we further divide each of $D_{m,c},DA_{m,c},DAR_{m,c}$ into $\left\lceil \frac{c}{m}\right\rceil$ segments, as $D_{m,c}=\left\{d_{m,m}^{1},\ldots ,d_{m,m}^{i},\ldots ,d_{m,m}^{\left\lceil \frac{c}{m}\right\rceil }\right\}$, $DA_{m,c}=\left\{da_{m,m}^{1},\ldots ,da_{m,m}^{i},\ldots ,da_{m,m}^{\left\lceil \frac{c}{m}\right\rceil }\right\}$ and $DAR_{m,c}=\left\{dar_{m,m}^{1},\ldots ,dar_{m,m}^{i},\ldots ,dar_{m,m}^{\left\lceil \frac{c}{m}\right\rceil }\right\}$, $1\leq i\leq \left\lceil \frac{c}{m}\right\rceil$, where the sub-relational features $d_{m,m}^{i},da_{m,m}^{i},dar_{m,m}^{i}\in R^{m\times m}$, where $1\leq i\leq \left\lceil \frac{c}{m}\right\rceil$.

To calculate the expected value of the distribution of the sub-relational features $d_{m,m}^{i},da_{m,m}^{i}$ and $dar_{m,m}^{i}$, where $1\leq i\leq \left\lceil \frac{c}{m}\right\rceil$, the eigenvectors $v_{d}^{i},v_{da}^{i}$, and $v_{dar}^{i}$, where $1\leq i\leq \left\lceil \frac{c}{m}\right\rceil$ are used to perform the determinant operation to obtain the directed area scalar of the eigenvector. In addition, the eigenvalue $\lambda_{d}^{i},\lambda_{da}^{i}$, and $\lambda_{dar}^{i}$, where $1\leq i\leq \left\lceil \frac{c}{m}\right\rceil$, are used to perform the operation of the expected value of the distribution. Therefore, we have
(14)$$\begin{array}{c} \left\{\begin{array}{c} d_{m,m}^{i}v_{d}^{i}=\lambda_{d}^{i}v_{d}^{i}\\ da_{m,m}^{i}v_{da}^{i}=\lambda_{da}^{i}v_{da}^{i}\\ dar_{m,m}^{i}v_{dar}^{i}=\lambda_{dar}^{i}v_{dar}^{i} \end{array}\right.,1\leq i\leq \left\lceil \frac{c}{m}\right\rceil . \end{array}$$

Finally, let $index_{i}$ be denoted as the index of the maximum expected value of the *i*-th segment, where $1\leq i\leq \left\lceil \frac{c}{m}\right\rceil$; the sub-relational features of each segment may be $d_{m,m}^{i},da_{m,m}^{i}$, and $dar_{m,m}^{i}$, based on their maximum expected value of the distribution, where $1\leq i\leq \left\lceil \frac{c}{m}\right\rceil$. Consequently, we have $index_{i}=max\left(\frac{det(v_{d}^{i})}{\lambda_{d}^{i}},\frac{det(v_{da}^{i})}{\lambda_{da}^{i}},\frac{det(v_{dar}^{i})}{\lambda_{dar}^{i}}\right)$, where $1\leq i\leq \left\lceil \frac{c}{m}\right\rceil$, by calculating the expected value of the distribution of $\frac{det(v_{d}^{i})}{\lambda_{d}^{i}},\frac{det(v_{da}^{i})}{\lambda_{da}^{i}}$, and $\frac{det(v_{dar}^{i})}{\lambda_{dar}^{i}}$. If $index_{i}$ is equal to $\frac{det(v_{d}^{i})}{\lambda_{d}^{i}}$, then *i*-segment is adopted $d_{m,m}^{i}$ by only considering the Doppler feature, where $1\leq i\leq \left\lceil \frac{c}{m}\right\rceil$. If $index_{i}$ is equal to $\frac{det(v_{da}^{i})}{\lambda_{da}^{i}}$, then *i*-segment adopts the $da_{m,m}^{i}$ by considering both the Doppler and angle features, where $1\leq i\leq \left\lceil \frac{c}{m}\right\rceil$. If $index_{i}$ is equal to $\frac{det(v_{dar}^{i})}{\lambda_{dar}^{i}}$, then *i*-segment is adopted the $dar_{m,m}^{i}$ by considering the Doppler, angle, and range features, where $1\leq i\leq \left\lceil \frac{c}{m}\right\rceil$. This work will be performed thorough the importance feature combination (IFC) operation, as defined below.

Then, the output of the Transformer-importance-decoder, $\bar{Y}$, can be represented, as given below. $\bar{Y}=\varepsilon^{t}\left\{IFC\left(d_{m,m}^{1},da_{m,m}^{1},dar_{m,m}^{1},index_{1}\right)⊕\cdots ⊕IFC\left(d_{m,m}^{i},da_{m,m}^{i},dar_{m,m}^{i},index_{i}\right)⊕\cdots ⊕IFC\left(d_{m,m}^{\left\lceil \frac{c}{m}\right\rceil },da_{m,m}^{\left\lceil \frac{c}{m}\right\rceil },dar_{m,m}^{\left\lceil \frac{c}{m}\right\rceil },index_{\left\lceil \frac{c}{m}\right\rceil }\right)\right\}$, where $\varepsilon^{t}$ is a parameter concatenation function to concatenate the training parameters by order through the features generated by iterative training, and updates the parameters obtained by the *t*-th decoding parameters from the previous (t−1)-th decoding parameters, where $\varepsilon^{t}=\varepsilon^{t-1}-\nabla L\left(Y,\bar{Y}\right)$. In addition, the importance feature combination (IFC) function, defined as the sub-relation features $d_{m,m}^{i},da_{m,m}^{i}$, or $dar_{m,m}^{i}$, is selected based on the value of $index_{i}$, where $1\leq i\leq \left\lceil \frac{c}{m}\right\rceil$. Figure 15 shows the example of the feature selection of the Transformer-importance-decoder. The loss function $L_{Imp}$ is calculated using Equation (Equation 3).

**Joint-decoding:** The CTC-decoder, Transformer-sequential-decoder, and Transformer-importance-decoder decode the predicted sequence-to-sequence results of $\hat{Y},\tilde{Y}$, and $\bar{Y}$, while *Y* is the ground-truth value of the sequence-to-sequence task. The best prediction sequence-to-sequence results of the CTC-decoder, Transformer-sequential-decoder, and Transformer-importance-decoder are ${\hat{Y}}^{*},{\tilde{Y}}^{*},{\bar{Y}}^{*}$ by iteratively searching for minimum values of $L_{CTC}^{i}\left(Y,\hat{Y}\right),L_{Seq}^{i}\left(Y,\tilde{Y}\right)$, and $L_{Imp}^{i}\left(Y,\tilde{Y}\right)$, for *R* times, where $L_{CTC}^{i},L_{Seq}^{i}$, and $L_{Imp}^{i}$ are the loss functions for *i*-th training for the CTC decoder, Transformer-sequential-decoder, and Transformer-importance-decoder, where 1≤i≤R, and *R* is the training number.
(15)$$\begin{array}{c} \left\{\begin{array}{c} {\hat{Y}}^{*}=\hat{Y}\;\mathrm{if}\;L_{\mathrm{CTC}}^{*}=\arg\;\min_{i=1}^{R}\;L_{\mathrm{CTC}}^{i}\left(Y,\hat{Y}\right),\;{\hat{Y}}^{*}=\hat{Y}\\ {\tilde{Y}}^{*}=\hat{Y}\;\mathrm{if}\;\;L_{\mathrm{Seq}}^{*}=\arg\;\min_{i=1}^{R}\;\;L_{\mathrm{Seq}}^{i}\left(Y,\tilde{Y}\right),\;\;{\tilde{Y}}^{*}=\tilde{Y}\\ {\bar{Y}}^{*}=\hat{Y}\;\mathrm{if}\;\;L_{\mathrm{Imp}}^{*}=\arg\;\min_{i=1}^{R}\;\;L_{\mathrm{Imp}}^{i}\left(Y,\bar{Y}\right),\;{\bar{Y}}^{*}=\bar{Y} \end{array}\right. \end{array}$$
where $L_{CTC}^{*}$, $L_{Seq}^{*}$, and $L_{Imp}^{*}$ are the final loss values of the CTC-decoder, Transformer-sequential-decoder, and Transformer-importance-decoder. After the obtaining the best sequence-to-sequence results of ${\hat{Y}}^{*}$, ${\tilde{Y}}^{*}$, ${\bar{Y}}^{*}$ of the CTC-decoder, Transformer-sequential-decoder, and Transformer-importance-decoder, the next purpose of the join-decoding operation is to search for the best training parameters α,β, and γ, as follows, to satisfy Equation (Equation 4).
(16)$$\begin{array}{c} \left\{\begin{array}{c} L^{*}=\arg\;\min_{\alpha ,\beta ,\gamma \in [0,1]}^{\alpha +\beta +\gamma =1}\;\alpha L_{CTC}^{*}+\beta L_{Seq}^{*}+\gamma L_{Imp}^{*}\\ Y^{*}=\alpha {\hat{Y}}^{*}+\beta {\tilde{Y}}^{*}+\gamma {\bar{Y}}^{*} \end{array}\right.\\ \mathrm{subject}\;\mathrm{to}\left\{\begin{array}{c} 0\leq \alpha \leq 1\\ 0\leq \beta \leq 1\\ 0\leq \gamma \leq 1\\ \alpha +\beta +\gamma =1 \end{array}\right. \end{array}$$

## 5. Experimental Result

In this section, the experimental setup is explained in Section 5.1, the CHR dataset is described in Section 5.2, and the performance evaluation is discussed in Section 5.3.

### 5.1. Experimental Setup

The environment setup is described, including the model parameter settings for our experimental results. In our experiment, the AI GPU utilizes NVIDIA RTX 3090, and the AI framework adopts TensorFlow 2.8 [29]. The programming environment is the Python 3.8 version under Windows 10. The mmWave sensor evaluation board is used AWR 1642 BOOST ODS [25] by Texas Instruments, which is a single-chip 76-GHz to 81-GHz automotive radar sensor integrating DSP and MCU, and the obstacle detection sensor includes onboard-etched antennas for the four receivers and two transmitters [25]. We also use DCA1000EVM [30] as the real-time data-capture adapter for the radar sensing evaluation module, as shown in Figure 16.

Our work is to investigate the continuous human motions, which include many single human motions. In our experiment, all of the single human motions include the wave, jumping forward, falling, lying down, walking forward, walking backward, walking left, walking right, running forward, running backward, running left, and running right. We additionally consider three different high-similarity motion groups; the walking group, running group, and vertical-motion group, among these single human motions. The single human motion of the walking group includes walking forward, walking backward, walking left, and walking right. The single human motion of the running group includes running forward, running backward, running left, and running right. The single human motion of the vertical-motion group includes falling and lying down. Note that each pair of single human motions from the same high-similarity motion groups is more difficult to distinguish that that of each pair of single human motions from the different high-similarity motion groups. We will further investigate the effects for our designed CHMR-HR scheme.

To see the effects of the radar sensing images with the capability of time-variable length, four different lengths are considered; there are 3, 6, 9, and 12 s. To further investigate the impact of radar sensing images with the time-variable length, we denoted a variable-time ratio = (1:1:1:1) for the same amount of training data sets of 3, 6, 9, and 12 s. To further understand the impacts of different variable-time ratios, we also consider the different variable-time ratios = (2:1:1:1), (1:2:1:1), (1:1:2:1), and (1:1:1:2) to increase the double amount of 3, 6, 9, and 12 s, respectively. In addition, each of the radar sensing images with time-variable lengths may randomly exist in one, two, three, or four different single human motions, in our experiment. The radar sensing data are collected using a mmWave sensor with onboard-etched MIMO antennas in a room with a length of four meters and a width of six meters, as shown in Figure 17. The experimental parameters are given in Table 1. To search for the best training parameters α, β, and γ for $L^{*}$ = arg min${}_{\alpha ,\beta ,\gamma \in [0,1]}^{\alpha +\beta +\gamma =1}\;\alpha L_{CTC}^{*}+\beta L_{Seq}^{*}+\gamma L_{Imp}^{*}$, we denote the training parameters ratio as (α,β,γ), where α,β,γ∈[0,1] and α+β+γ=1, in our experiment.

### 5.2. The CHR Dataset

The dataset of continuous human motions for using AWR 1642 BOOST ODS and DCA1000EVM [25,30] is illustrated in Figure 17. We collected a training dataset of 1700 records: 500 records for the 3 s data, 426 records for the 6 s data, 416 records for the 9 s data, and 358 records for the 12 s data. Additionally, we collected a test dataset of 500 records, with 160 records for the 3 s data, 148 records for the 6 s data, 110 records for the 9 s data, and 82 records for the 12 s data. All these datasets were collected by 10 participants in our laboratory. There were five men and five women. They were 22–28 years old, 45–85 kg in weight, and 155–183 cm in height. To see the effect of data from the different backgrounds, we collected data from two different rooms with the different environments. We collected 80% raw data from a big space room without a complicated background, and we also collected 20% raw data from the another small space room with a more complicated background.

### 5.3. Performance Analysis

This subsection discussed the performance analysis. In our experimental, the real sample is the data that the trainer manually labels for each CHM radar data, the positive samples are the CHM radar data whose label is consistent with the real sample, and the negative sample are the CHM radar data whose label is inconsistent with the real sample.

We use the True Positive (TP), True Negative (TN), False Positive (FP), and False Negative (FN) for our experiment, as follows. In our work, True Positive (TP) denoted that the predicted result and the labels of continuous motion in the positive sample are the same as the correct label prediction, in the positive samples of the CHM dataset. True Negative (TN) denoted that the predicted results and the continuous motions labels in the negative samples are both wrong label predictions in the negative samples of the CHM dataset. False Positive (FP) denoted that the predicted result is different from the wrong label in the negative sample; the model’s prediction is the result of the correct label in the negative samples of the CHM dataset. False Negative (FN) denoted that the predicted result is different from the correct label in the positive sample, and the model’s prediction is the result of the wrong label in the positive samples of the CHM dataset.

The performance metrics of our experimental to be observed are:(1)*Accuracy (AC)* is the value of $1-L^{*}$, and $L^{*}$ = arg min${}_{\alpha ,\beta ,\gamma \in [0,1]}^{\alpha +\beta +\gamma =1}\;\alpha L_{CTC}^{*}+\beta L_{Seq}^{*}+\gamma L_{Imp}^{*}$, where $L_{CTC}^{*},L_{Seq}$, and $L_{Imp}^{*}$ are the final loss values of CTC-decoder, Transformer-sequential-decoder, and Transformer-importance-decoder, under the real samples, including positive and negative samples, are considered for the test dataset.(2)*Recall (RE)* is True Positive (TP) divided by the sum of True Positive (TP) with False Negative (FN), i.e., $\frac{\mathrm{TP}}{\mathrm{TP}+\mathrm{FN}}$.(3)*Precision (PC)* is True Positive (TP) divided by the sum of True Positive (TP) with False Positive (FP), i.e., $\frac{TP}{\mathrm{TP}+\mathrm{FP}}$.(4)*F1-score (F1)* is twice the Precision (PC) times Recall (RE) divided by the sum of Precision (PC) with Recall (RE), i.e., $\frac{2\times \mathrm{PC}\times \mathrm{RE}}{\mathrm{PC}+\mathrm{RE}}$.

An efficient CHMR-HS scheme using mmWave radar sensing data is achieved with a high AC, high RE, high PC, and high F1. Efforts will be made in this paper to improve the AC, RE, PC, and F1.

#### 5.3.1. Accuracy (AC)

Before discussing with the accuracy, we must determine the best training parameters α,β, and γ for $L^{*}$ = arg min${}_{\alpha ,\beta ,\gamma \in [0,1]}^{\alpha +\beta +\gamma =1}\;\alpha L_{CTC}^{*}+\beta L_{Seq}^{*}+\gamma L_{Imp}^{*}$, such that the training parameter ratio = (α,β,γ), where α,β,γ∈[0,1] and α+β+γ=1, can be determined. To show how to determine the best training parameter ratio (α,β,γ), nine different training parameter ratios (α,β,γ) are set as (0.8, 0.1, 0.1), (0.1, 0.8, 0.1), (0.1, 0.1, 0.8), (0.6, 0.2, 0.2), (0.2, 0.6, 0.2), (0.2, 0.2, 0.6), (0.4, 0.3, 0.3), (0.3, 0.4, 0.4), and (0.3, 0.3, 0.4). We apply these different values of (α,β,γ) to calculate the accuracy, and we observed that an accuracy = 94.5% is obtained if (α,β,γ) = (0.3,0.4,0.3), which is higher than those of other values of (α,β,γ) if 2D data are used, as shown in Figure 18a. Similarly, we also observed that an accuracy = 95.2% is obtained if (α,β,γ) = (0.3,0.3,0.4), which is higher than those of other values of (α,β,γ) if the 3D data are used, as shown in Figure 18b. Therefore, the best training parameters (α,β,γ) = (0.3,0.4,0.3) and (α,β,γ) = (0.3,0.3,0.4) are used for the 2D and 3D data, for the following discussions.

To make a fair comparison, the proposed CHMR-HS scheme for the 3/2D features (using red lines with square and circle marks), the proposed CHMR-HS scheme without CTC for 3/2D features (using red lines with cross-box and cross-circle marks), the CTC scheme for 3/2D features (using red lines with hollow-box and hollow-circle marks), and a CHMR scheme is proposed by Zhao et al. [11], simplified as the Zhao-I scheme (using a blue line). An end-to-end network for CHMR is proposed by Zhao et al. [1], simplified as the Zhao-II scheme (green line), and another CHMR scheme is presented by Shrestha et al. [10], simplified as the Shrestha scheme (orange line) are compared in our experiment.

The experimental results of accuracy (AC) vs. epoch are shown in Figure 19. As shown in Figure 19, we observed that the accuracy (AC) of the CHMR-HS scheme with 3D features (red line with square mark) = 95.2% > that of the CHMR-HS scheme with 2D features (red line with circle mark) = 94.5% > that of the CHMR-HS scheme without CTC for the 3D features (red line with cross-box mark) = 94.1% > that of the CHMR-HS scheme without CTC for 2D features (red line with cross-circle mark) = 93.4% > that of the Zhao-I scheme (blue line) = 93.1% > that of the Zhao-II scheme (green line) = 92.7% > that of the Shrestha scheme (orange line) = 92.1% > that of the CTC scheme for 3D features (red line with hollow-box mark) = 91.9% > that of the CTC scheme for 2D features (red line with hollow-circle mark) = 91.3% after the epoch is equal to 300. In addition, we also observed that the AC convergence rate of the CHMR-HS scheme with 3D features is lower than that of the CHMR-HS scheme with 2D features, but the final accuracy of the CHMR-HS scheme with 3D features is 95.2%, which is higher than that of the CHMR-HS scheme with 2D features, which is 94.5%.

To investigate the impacts of different number of features, we perform our CHMR-HR under a given different number of 3D features and 2D features. As shown in Figure 20, we observed that the accuracy of the CHMR-HS scheme with RDS + RAS + RES = 95.2% > that of CHMR-HS scheme with DS + AS + ES + RS = 94.5% > that of CHMR-HS scheme with RDS = 93.6% > that of CHMR-HS scheme with DS + AS + RS = 94% > that of CHMR-HS scheme with DS = 92.9%. Basically, the more features that are utilized for training, the higher the accuracy will be.

To investigate the ability of the proposed method to recognize different motions under different durations, we used 3 s, 6 s, 9 s, 12 s continuous motion samples directly collected in a real environment for testing. The example timeline of the proposed method and the compared methods is shown in Figure 21 for recognizing walking forward, walking right, running back, and falling, where the recognition results generated by each method are recorded along the time axis. It can be seen that misjudgments of CHMR-HS with the 3/2D features mostly occur at the time frames of motion transitions due to the use of the proposed Transformer-sequential-decoder and the Transformer-importance-decoder. Moreover, the Zhao-I, Zhao-II, and Shrestha schemes have high false recognition rates. This verifies that the proposed method illustrates that a better recognition performance is achieved.

In the following Tables, bold and underlined numbers are denoted as as the first and second places of our simulation results. All motions denoted as the motions of CHMR randomly comes from all groups; the walking group, running group, and vertical-motion group denoted as the motions of CHMR come from the walking group, running group, and vertical-motion group, respectively. As shown in Table 2, the accuracy of the CHMR-HS scheme with 3D features are always better than that of the CHMR-HS scheme with 2D features, except for the vertical-motion group. This is because that the classification of the vertical-motion group can be influenced by the oscillation of the characteristic frequency of 2D features, so that the learning model can more easily learn the difference of the acceleration gap between different motions in a vertical highly similar motion.

#### 5.3.2. Recall (RE)

The experimental results of recall (RE) vs. epoch are shown in Figure 22. As shown in Figure 22, we observed that the recall (RE) of the CHMR-HS scheme with 3D features (RDS + RAS + RES) = 93.7% > that of the CHMR-HS scheme with 2D features (DS + AS + ES + RS) = 92.2% > that of CHMR-HS scheme without CTC for 3D features (RDS + RAS + RES) = 91.9% > that of the CHMR-HS scheme without CTC for 2D features (DS + AS + ES + RS) = 91.2% > that of Zhao-I scheme (DS) = 90.9% > that of Zhao-II scheme (DS) = 90.1% > that of CTC scheme for 3D features (RDS + RAS + RES) = 87.9% > that of Shrestha scheme (DS) = 88.2% > that of CTC scheme for 2D features (DS + AS + ES + RS) = 87.9% if the epoch is 300. In addition, we also observed that the RE convergence rate of the CHMR-HS scheme with 3D features is lower than that of the CHMR-HS scheme with 2D features.

As shown in Table 3, the Recall (RE) of the CHMR-HS scheme with 3D features is also always better than that of the CHMR-HS scheme with 2D features, except for the vertical-motion group, due to the classification of vertical-motion group being more easily influenced by the oscillation of the characteristic frequencies of 2D features.

#### 5.3.3. Precision (PR)

The experimental results of precision (PR) vs. epoch are shown in Figure 23. As shown in Figure 23, we observed that the precision (PR) of the CHMR-HS scheme with 3D features (RDS + RAS + RES) = 84.8% > that of the CHMR-HS scheme without CTC for 3D features (RDS + RAS + RES) = 82.8% > that of CHMR-HS scheme with 2D features (DS + AS + ES + RS) = 81.9% > that of the Zhao-I scheme (DS) = 81.8% > that of the Zhao-II scheme (DS) = 79.1% > that of the CHMR-HS scheme without CTC for 2D features (DS + AS + ES + RS) = 78.5% > that of the CTC scheme for 3D features (RDS + RAS + RES) = 78.9% > that of the CTC scheme for 2D features (DS + AS + ES + RS) = 78.5% > that of the Shrestha scheme (DS) = 77.1% if the epoch is 300. In addition, we also observed that the RR convergence rate of the CHMR-HS scheme with 3D features is lower than that of CHMR-HS scheme with 2D features.

We observed that the performance of recall (RE) of our CHMR-HS scheme with 2D features is higher than that of the CHMR-HS scheme without CTC-decoder, but the precision (PC) of our CHMR-HS scheme with 2D features is lower than that of the CHMR-HS scheme without CTC, where RE is $\frac{\mathrm{TP}}{\mathrm{TP}+\mathrm{FN}}$, and PC is $\frac{\mathrm{TP}}{\mathrm{TP}+\mathrm{FP}}$. This is because the prediction of the False Positive (FP) of the CHMR-HS scheme without CTC-decoder will be better than the prediction of the False Negative (FN), due to the CTC-decoder being too optimistic to predict for the positive samples.

As shown in Table 4, the precision (PR) of the CHMR-HS scheme with 3D features are also always better than that of CHMR-HS scheme with 2D features, except for the vertical-motion group.

#### 5.3.4. F1-Score (F1)

The experimental results of the F1-score (F1) vs. epoch are shown in Figure 23. As shown in Figure 24, we observed that the F1-score (F1) of the CHMR-HS scheme with 3D features (RDS + RAS + RES) = 89% > that of the CHMR-HS scheme without CTC for 3D features (RDS + RAS + RES) = 87.1% > that of the CHMR-HS scheme with 2D features (DS + AS + ES + RS) = 86.7% > that of the Zhao-I scheme (DS) = 85.9% > that of the CHMR-HS scheme without CTC for 2D features (DS + AS + ES + RS) = 85.8% > that of the Zhao-II scheme (DS) = 84.2% > that of the CTC scheme for 3D features (RDS + RAS + RES) = 83.5% > that of the CTC scheme for 2D features (DS + AS + ES + RS) = 82.9% > that of the Shrestha scheme (DS) = 82.2% if the epoch is 300. In addition, we also observed that the RR convergence rate of the CHMR-HS scheme with 3D features is lower than that of the CHMR-HS scheme with 2D features.

We observed that the performance of recall (RE) of our CHMR-HS scheme with 2D features is higher than that of the CHMR-HS scheme without CTC, but the F1-score (F1) of our CHMR-HS scheme with 2D features is also lower than that of CHMR-HS scheme without CTC, due to the precision (PC) of our CHMR-HS scheme with 2D features being lower than that of the CHMR-HS scheme without CTC, where the F1-score (F1) function is $\frac{2\times \mathrm{PC}\times \mathrm{RE}}{\mathrm{PC}+\mathrm{RE}}$.

As shown in Table 5, the F1-score (F1) of the CHMR-HS scheme with 3D features are also always better than that of the CHMR-HS scheme with 2D features, except for the vertical-motion group.

#### 5.3.5. Discussion on High Similarity

We investigate the ability of the proposed method to recognize different motions under different durations. We used 3 s, 6 s, 9 s, and 12 s continuous motion samples directly collected in a real environment for testing. To discuss the impact for the continuous human motions with high similarity, we try to independently increase in proportion for 3 s, 6 s, 9 s, 12 s; i.e., variable-time ratio = (1:1:1:1), (2:1:1:1), (1:2:1:1), (1:1:2:1), or (1:1:1:2) are considered.

Table 6 illustrates that the performance comparison of all motions and walking, running, and vertical-motion groups, if using all data of the training dataset. In Table 6, all of the performance results of accuracy, recall, precision, and F1-score of CHMR-HS with 3D features are better than those of CHMR-HS with 2D features, except for the vertical-motion group. The reason for this that the classification of the vertical-motion group can be more easily influenced by the oscillation of characteristic frequency if using 2D features.

Table 7 illustrates that the performance comparisons of all motions under variable-time ratios = (2:1:1:1), (1:2:1:1), (1:1:2:1), and (1:1:1:2). To consider all the motions, we observed that the performance results of accuracy, recall, precision, and F1-score of CHMR-HS with 2D features are better than that of CHMR-HS with 3D features if variable-time ratios = (2:1:1:1) or (1:2:1:1) are used. However, the performance results of accuracy, recall, precision, and F1-score of CHMR-HS with 3D features are better than that of CHMR-HS with 2D features if the variable-time ratios = (1:1:2:1), and (1:1:1:2) are used. This is because the 3D features require a longer data length to learn the dependencies of continuous motions.

Table 8 illustrates the performance comparisons of the walking group under variable-time ratios = (2:1:1:1), (1:2:1:1), (1:1:2:1), and (1:1:1:2). To consider the CHM from the walking group, we observed that the performance results of accuracy, recall, precision, and F1-score of CHMR-HS with 3D features are better than those of CHMR-HS with 2D features for all variable-time ratios = (2:1:1:1), (1:2:1:1), (1:1:2:1), or (1:1:1:2) used.

As compared with the performance result of the walking group in Table 6, we observed that the better performance results of the walking group are obtained if the variable-time ratios = (2:1:1:1) or (1:2:1:1) are used, but the lower performance results of walking group will occur if the variable-time ratios = (1:1:2:1) or (1:1:1:2) are used. This is because the training data under the variable-time ratios (2:1:1:1) and (1:2:1:1) are provided to learn the periodic characteristics of the micro-vibrations of the walking motions, such that the learning model can be strengthened to identify micro-differences between the walking motions through the feature performance of a short period of time of training data.

Table 9 illustrates that the performance comparisons of the running group under variable-time ratios = (2:1:1:1), (1:2:1:1), (1:1:2:1), and (1:1:1:2). To consider the CHM from the running group, we observed that the performance results of accuracy, recall, precision, and F1-score of CHMR-HS with 3D features are better than those of CHMR-HS with 2D features for all variable-time ratios = (2:1:1:1), (1:2:1:1), (1:1:2:1), or (1:1:1:2) are used.

As compared with the performance result of the running group in Table 6, we observed that the better performance results of the running group are obtained if variable-time ratios = (2:1:1:1) or (1:2:1:1) are utilized, but the lower performance results of the running group will be if variable-time ratios = (1:1:2:1) or (1:1:1:2) are utilized. This is because the training data under the variable-time ratios (2:1:1:1) and (1:2:1:1) are provided to learn the periodic characteristics of the micro-vibrations of the running motions, such that the learning model can be strengthened to identify micro-differences between running motions through the feature performance of a short period of time of training data.

Table 10 illustrates the performance comparisons of the vertical-motion group under variable-time ratios = (2:1:1:1), (1:2:1:1), (1:1:2:1), and (1:1:1:2). To consider the CHM from the vertical-motion group, we observed that the performance results of accuracy, recall, precision, and F1-score of CHMR-HS with 2D features are better than those of CHMR-HS with 3D features for all variable-time ratios = (2:1:1:1), (1:2:1:1), (1:1:2:1), or (1:1:1:2) are used.

As compared with the performance results of the vertical-motion group in Table 6, we observed that the better performance results of the vertical-motion group are obtained if variable-time ratios = (2:1:1:1) or (1:2:1:1) are utilized, but the lower performance results of the vertical-motion group will occur if variable-time ratios = (1:1:2:1) or (1:1:1:2) are utilized. This is because that the training data under the variable-time ratios (2:1:1:1) and (1:2:1:1) are provided to learn the periodic characteristics of the micro-vibrations of the vertical-motion, such that the learning model can be strengthened to identify micro-differences between vertical-motions through the feature performance of a short period of time of training data.

## 6. Conclusions

In this paper, we proposed a fully new CHMR-HS scheme for continuous human motion recognition using mmWave FMCW radar, to specially consider the continuous human motion recognition with a high-similarity problem. A Transformer-sequential-based learning mechanism is re-designed from the traditional Transformer-encoder–decoder architecture. We also adopt the 2D/3D features extracted from mmWave FMCW radar images as the input of our Transformer-sequential-based learning model. The experimental results show that the accuracies of our proposed CHMR-HS scheme scheme are 95.2% and 94.5%, without considering the high-similarity condition, higher than those of existing CHMR schemes, and the accuracies of our proposed CHMR-HS scheme are 94.4% and 94.1%, especially considering the high-similarity condition. The results illustrate that our proposed CHMR-HS scheme can significantly improve the accuracy of continuous human motions, even for the high-similarity condition.

In future works, we will make research of the human motion recognition for multi-model cross learning, by combining the FMCW radar images with camera RGB images to increase the recognition accuracy. We also conduct further research toward continuous human motion recognition and vital signal measurements through FMCW radar for the multiple moving targets under deploying distributed radars.

## Figures and Tables

**Figure 1 sensors-22-08409-f001:**
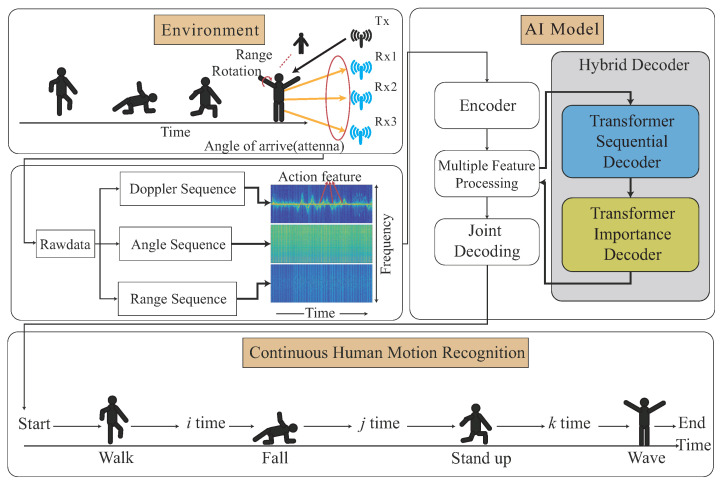
Our proposed CHMR-HS scheme.

**Figure 2 sensors-22-08409-f002:**
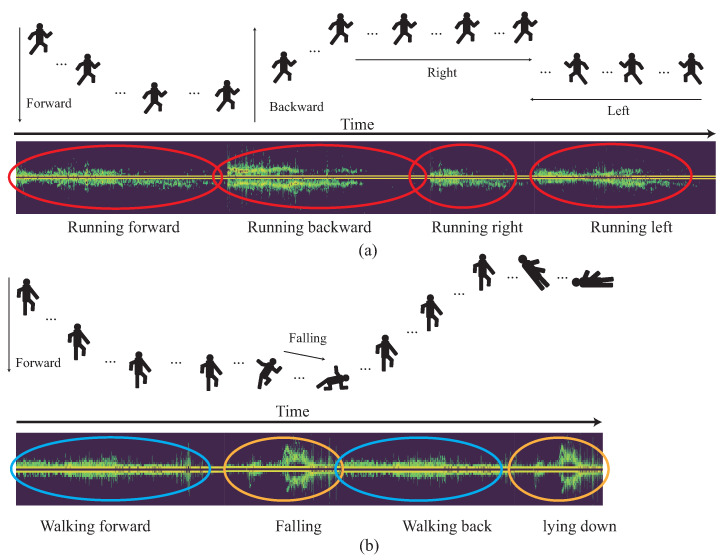
Schematic diagram of continuous human motion with high similarity; (**a**) continuous motion of running motion group, (**b**) continuous motion of walking, lying down, and falling motions.

**Figure 3 sensors-22-08409-f003:**
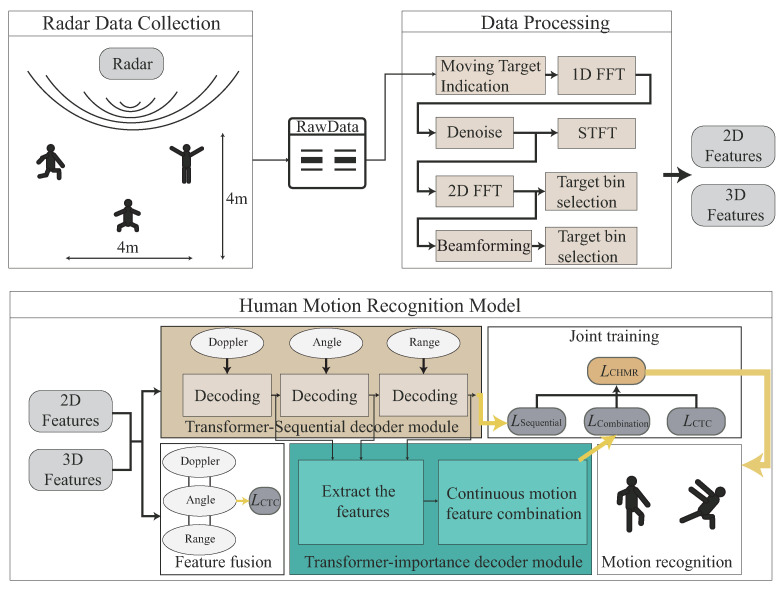
System structure of CHMR-HS scheme.

**Figure 4 sensors-22-08409-f004:**
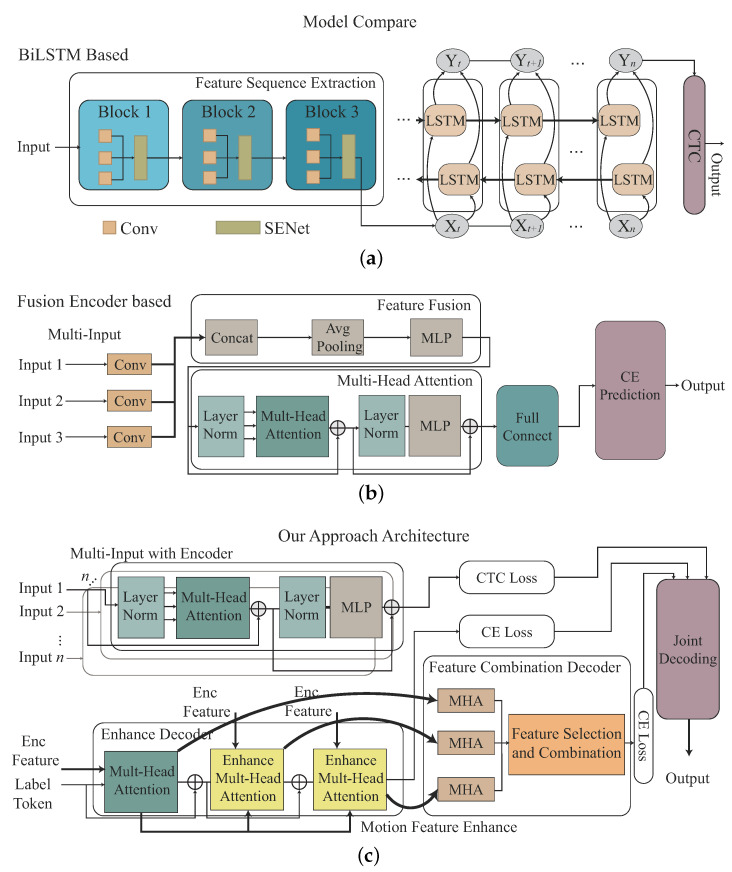
Comparison with (**a**) the MSENet BLSTM scheme, (**b**) Transformer-Encoder scheme, and (**c**) our proposed CHMR-HS scheme.

**Figure 5 sensors-22-08409-f005:**
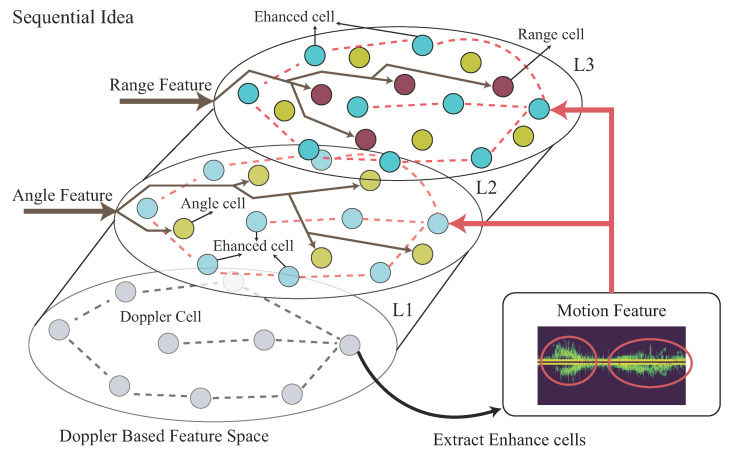
Basic idea of Transformer-sequential-decoder scheme.

**Figure 6 sensors-22-08409-f006:**

The flowchart of the proposed CHMR-HS scheme.

**Figure 7 sensors-22-08409-f007:**
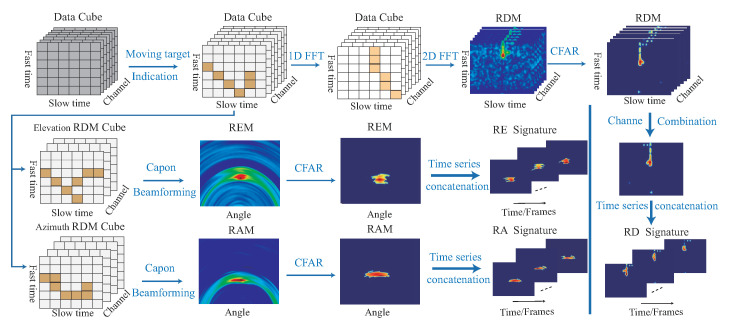
Example of range-Doppler signature $RDS_{T}$, range-azimuth signature $RAS_{T}$, and range-elevation signature $RES_{T}$.

**Figure 8 sensors-22-08409-f008:**
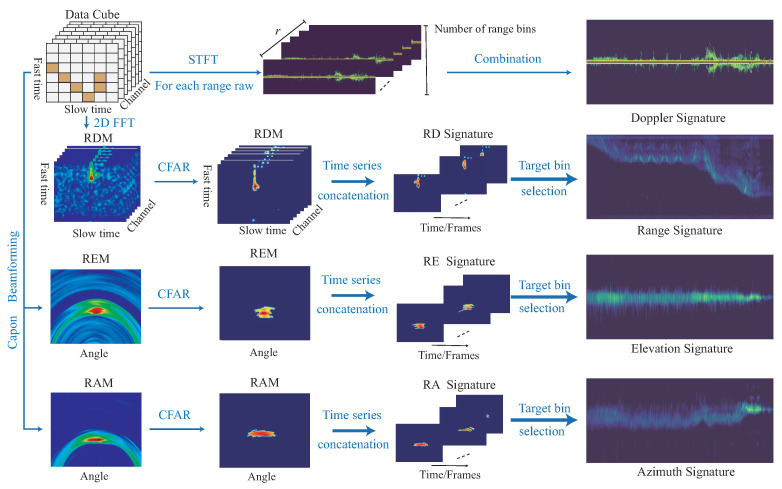
Example of the generation of 2D feature signatures.

**Figure 9 sensors-22-08409-f009:**
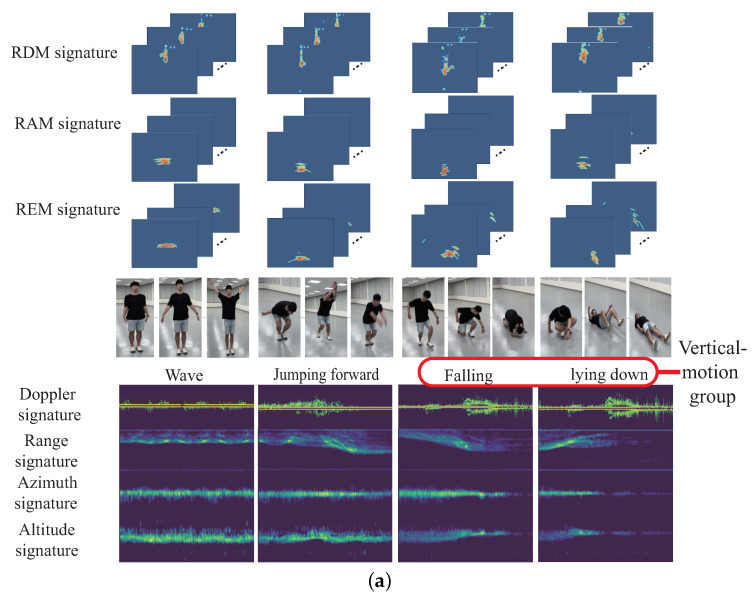
(**a**) Human motions of wave, jumping forward, vertical-motion group (including falling and lying down), (**b**) human motions of walking group (including walking forward, walking backward, walking left, walking right), (**c**) human motions of running group (including running forward, running backward, running left, running right).

**Figure 10 sensors-22-08409-f010:**
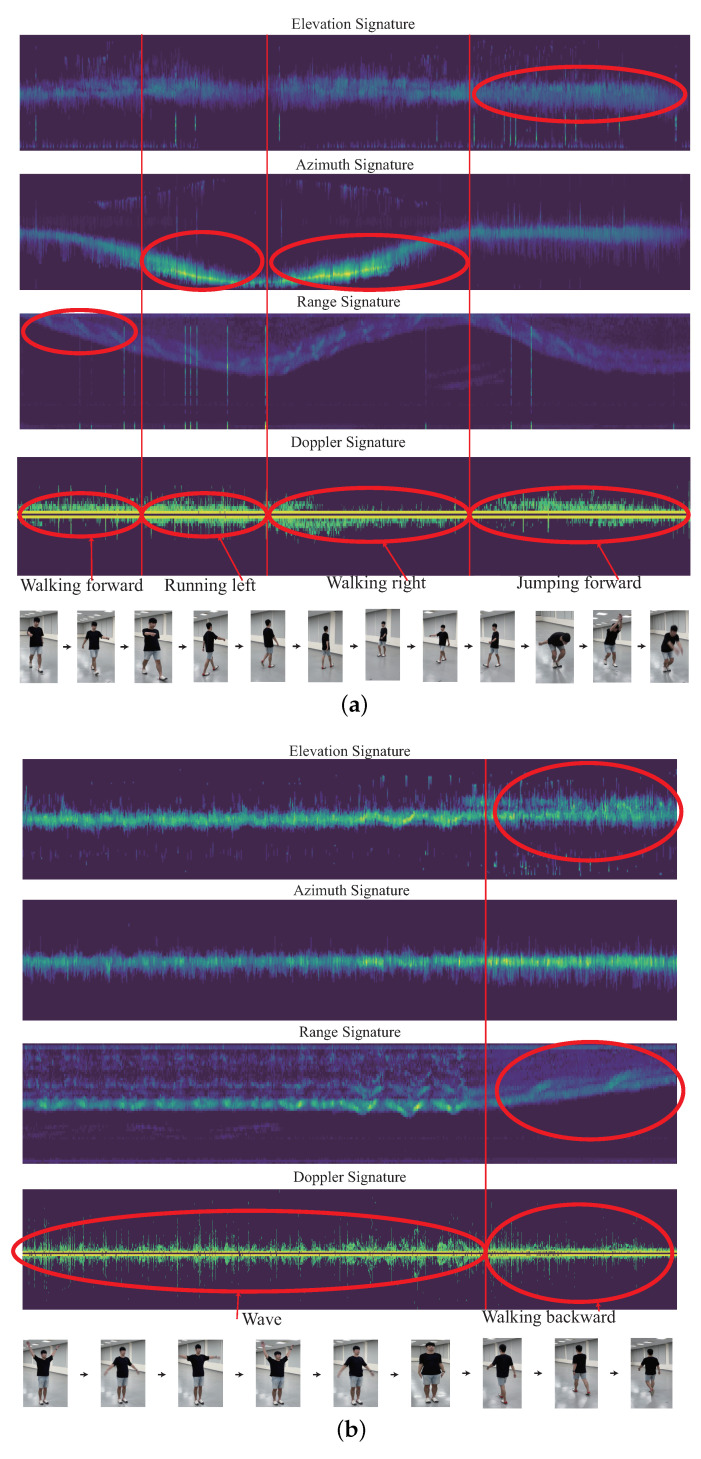
Examples of continuous human motions with 2D feature signatures under (**a**) four motions (12 s), (**b**) two motions (12 s).

**Figure 11 sensors-22-08409-f011:**
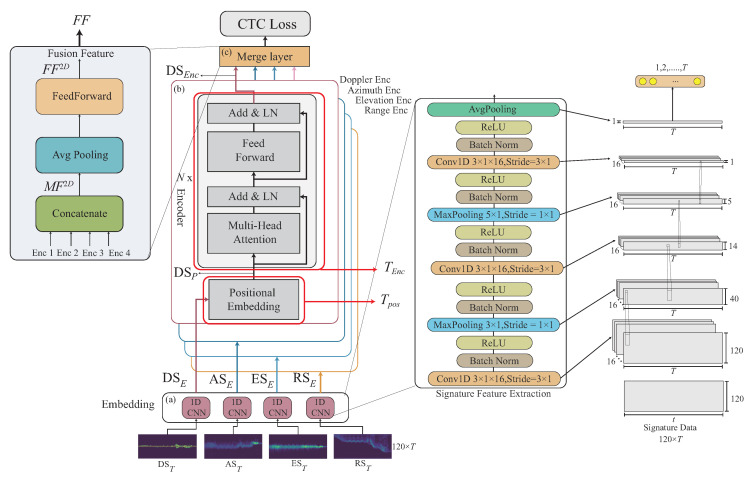
2D Transformer-based encoder with (**a**) CNN embedding module, (**b**) multi-head attention module, and (**c**) fusion feature module.

**Figure 12 sensors-22-08409-f012:**
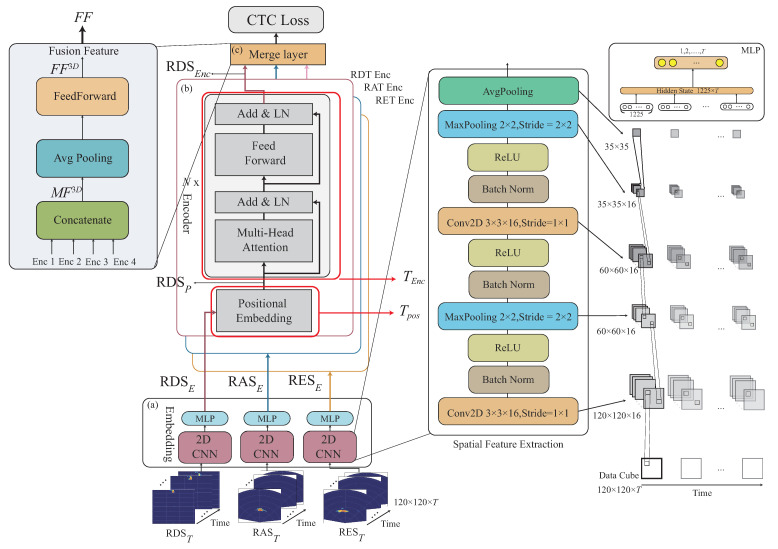
3D Transformer-based encoder with (**a**) CNN embedding module, (**b**) multi-head attention module, and (**c**) fusion feature module.

**Figure 13 sensors-22-08409-f013:**
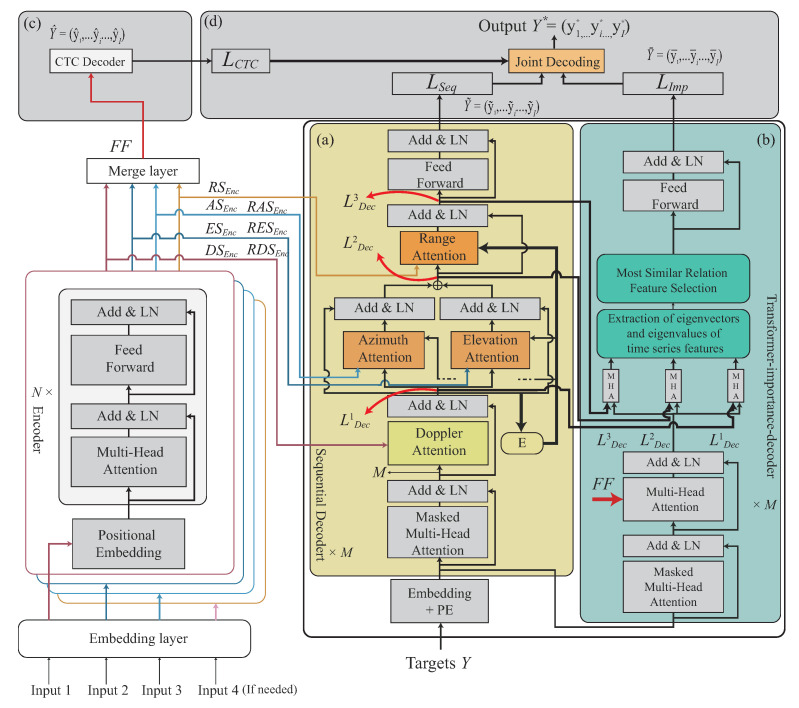
Our CHMR-HS architecture with (**a**) a Transformer-sequential-decoder module, (**b**) a Transformer-importance-decoder module, (**c**) CTC decoder, and (**d**) joint decoding module.

**Figure 14 sensors-22-08409-f014:**
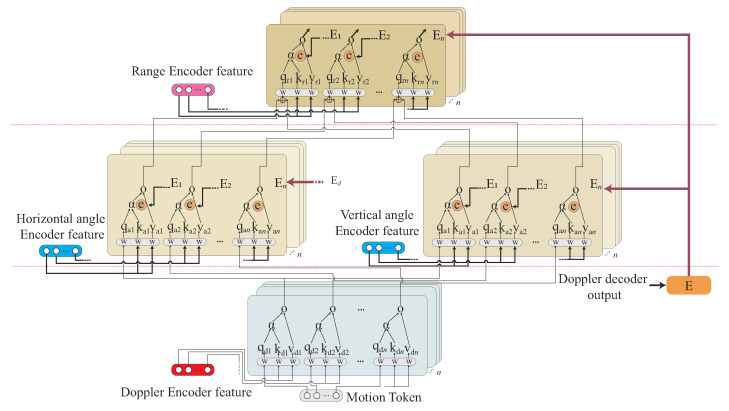
Transformer-sequential-decoder for sequentially decoding the Doppler, Doppler- angle, and Doppler-angle- range features.

**Figure 15 sensors-22-08409-f015:**
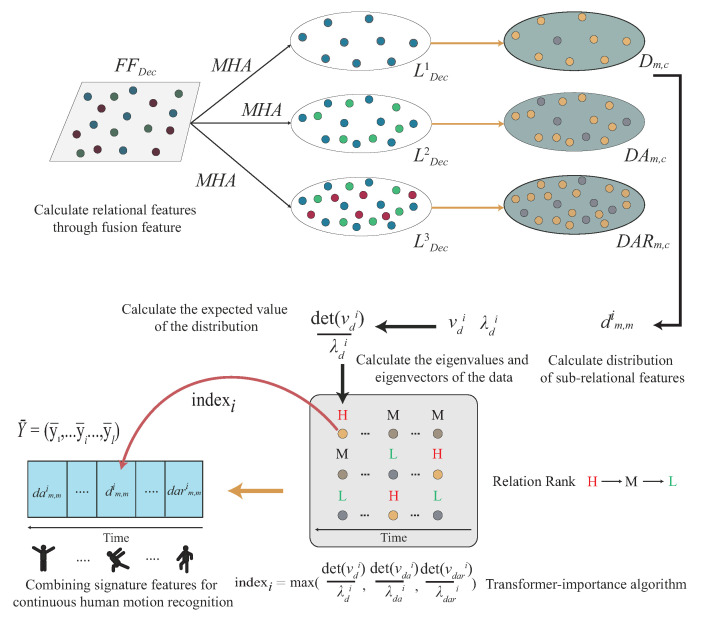
The feature selection of the Transformer-importance-decoder.

**Figure 16 sensors-22-08409-f016:**
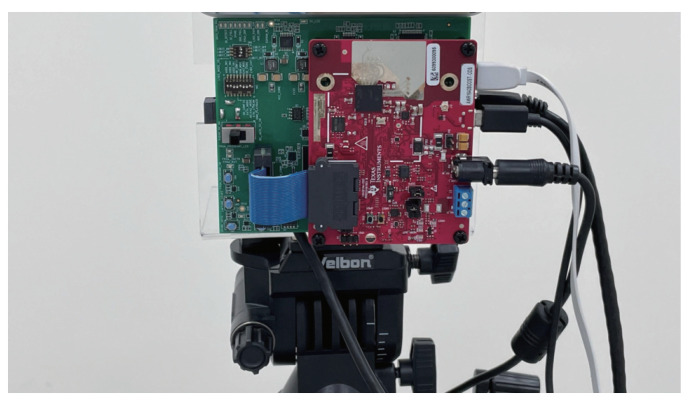
The hardware of mmWave sensor evaluation board and data-capture adapter.

**Figure 17 sensors-22-08409-f017:**
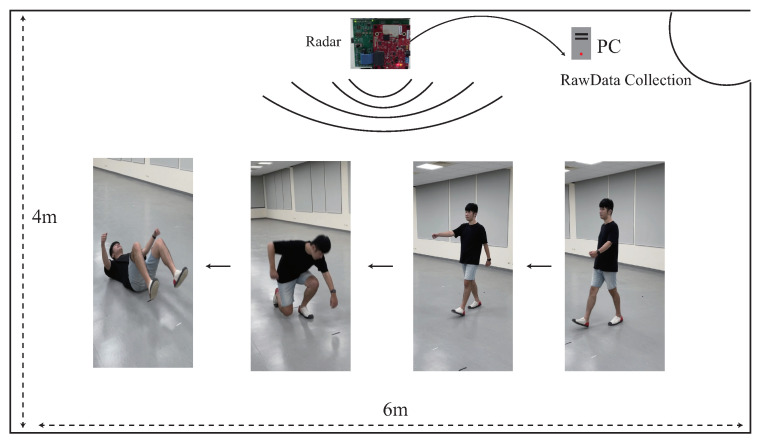
Environment of the experimental setup with radar antennas.

**Figure 18 sensors-22-08409-f018:**
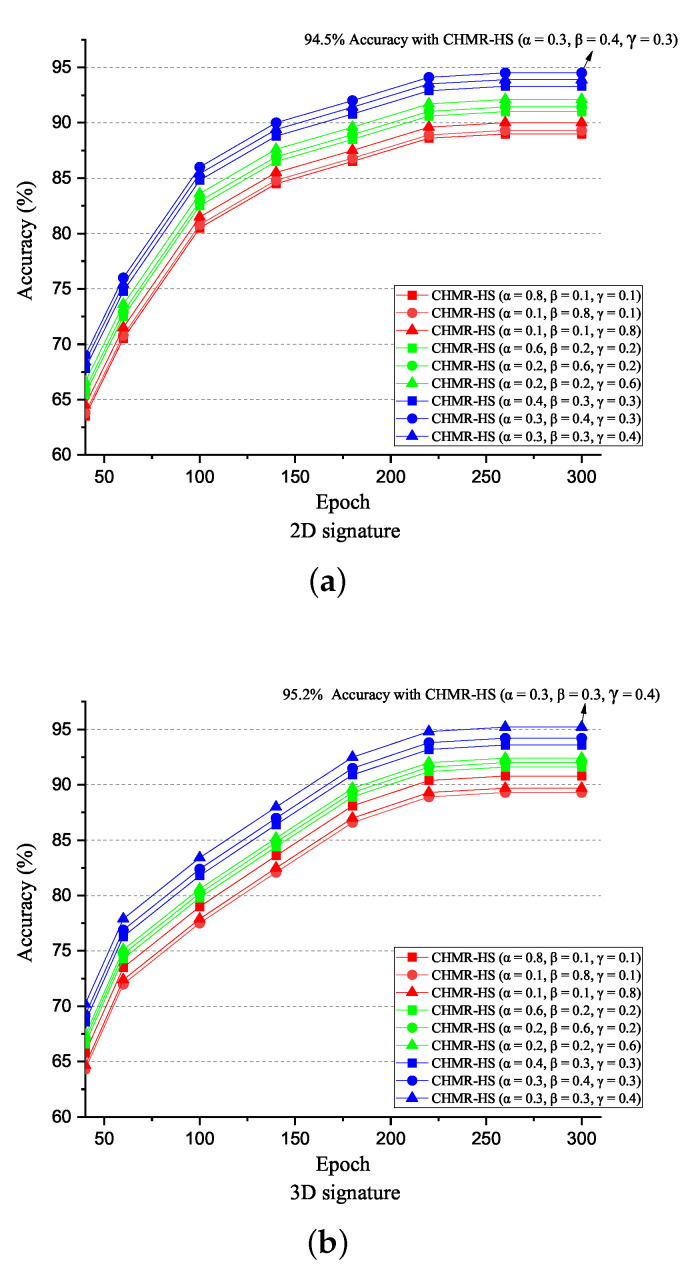
The performance of accuracy if (α,β,γ) = (0.8,0.1,0.1), (0.1,0.8,0.1), (0.1,0.1,0.8), (0.6,0.2,0.2), (0.2,0.6,0.2), (0.2,0.2,0.6), (0.4,0.3,0.3), (0.3,0.4,0.4) and (0.3,0.3,0.4), for (**a**) 2D data, and (**b**) 3D data.

**Figure 19 sensors-22-08409-f019:**
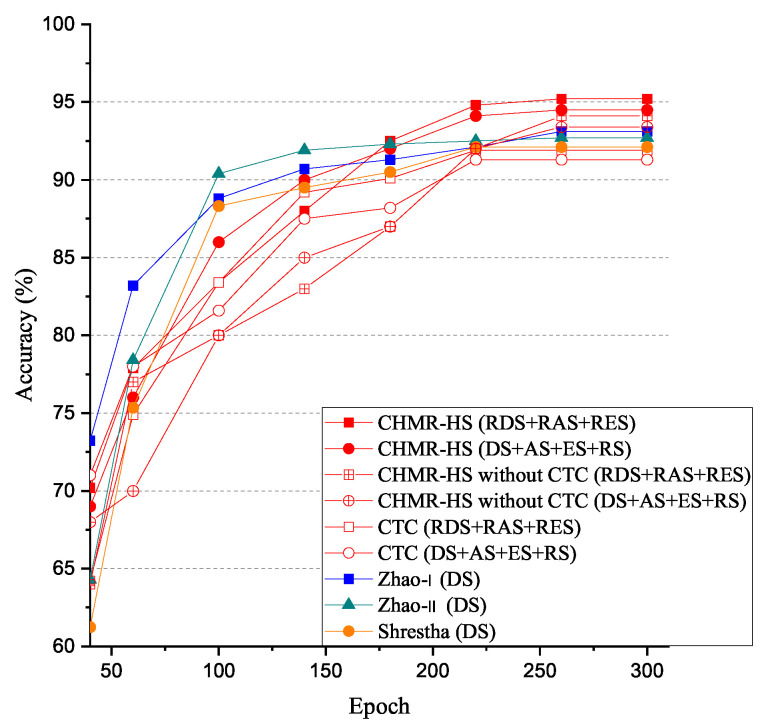
The performance of accuracy (AC) vs. epoch.

**Figure 20 sensors-22-08409-f020:**
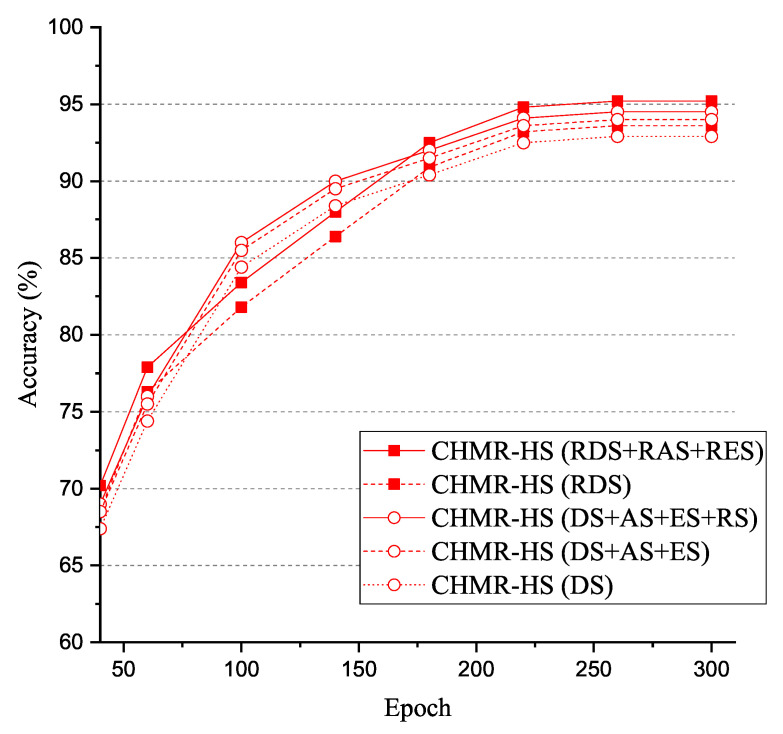
Performance of accuracy (AC) vs. epoch under different number of features.

**Figure 21 sensors-22-08409-f021:**
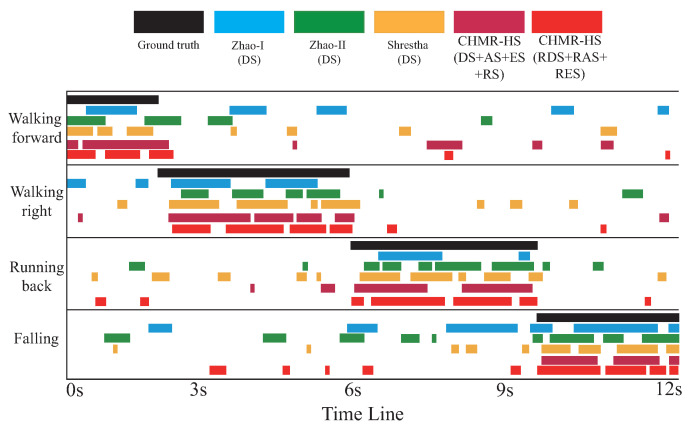
Example timeline of different CHMR methods for recognizing walking forward, walking right, running back, and falling.

**Figure 22 sensors-22-08409-f022:**
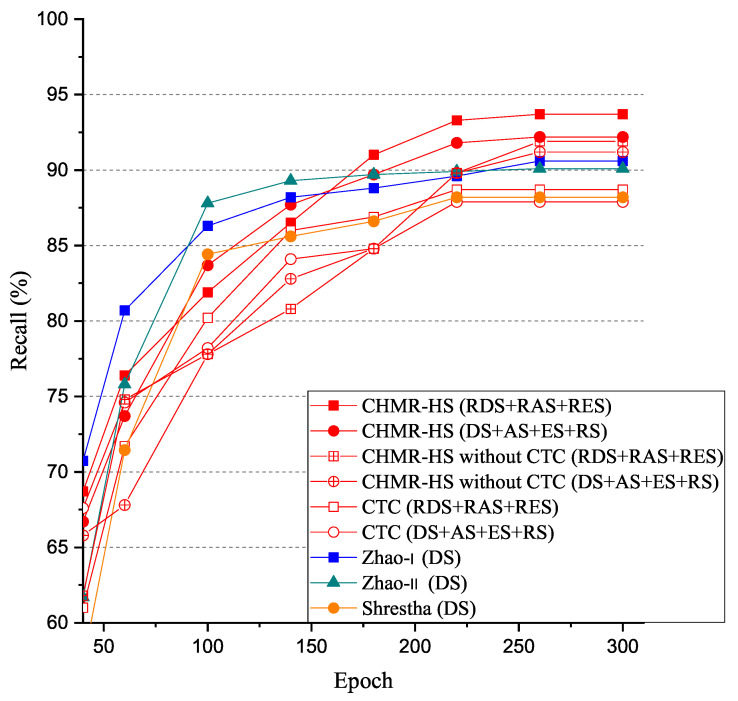
The performance of recall (RE) vs. epoch.

**Figure 23 sensors-22-08409-f023:**
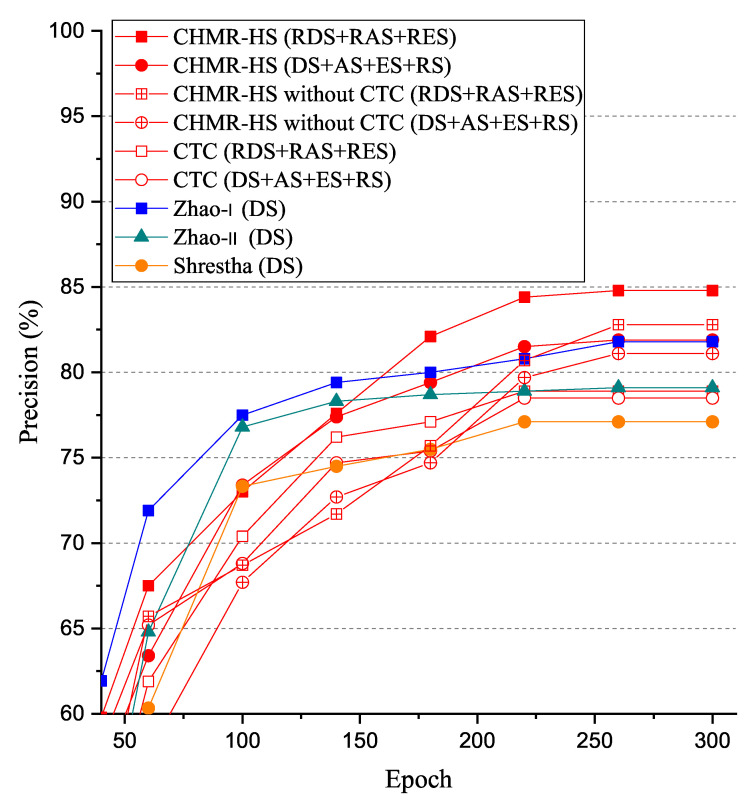
The performance of precision (PR) vs. epoch.

**Figure 24 sensors-22-08409-f024:**
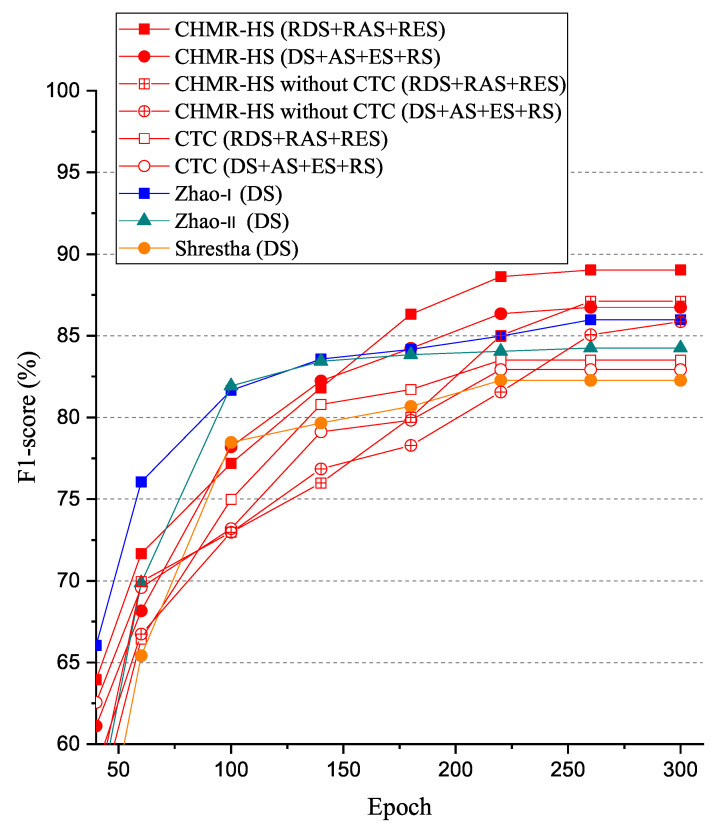
The performance of F1-score (F1) vs. epoch.

**Table 1 sensors-22-08409-t001:** Experiment parameter.

Radar	AWR 1642BOOST ODS
Start Frequency	77 Ghz
End Frequency	81 Ghz
Number of Sample	120
Number of Chirp	120
Number of Tx antennas	2
Number of Rx antennas	4
Learning rate	0.001
Drop out	0.5
Number of encoder	6
Number of decoder	6

**Table 2 sensors-22-08409-t002:** The comparison of accuracy (AC) for all motions and walking, running, and vertical-motion groups.

Data Type	Models	Accuracy			
		All Motions	Walking Group	Running Group	Vertical-Motion Group
**2D Features**	CHMR-HS (DS+AS+ES+RS)	94.5%	94.6%	92.9%	**95%**
	CHMR-HS without CTC (DS+AS+ES+RS)	93.4%	84.5%	86.1%	89.3%
	Zhao-I (DS)	93.1%	82.55%	79.2%	88.25%
	Zhao-II (DS)	92.7%	83.34%	81.9%	83.37%
	Shrestha (DS)	92.1%	79.3%	83.33%	83.56%
	CTC (DS+AS+ES+RS)	91.3%	81.6%	83.5%	86.2%
**3D Features**	CHMR-HS (RDS+RAS+RES)	**95.2%**	**95.3%**	**93.8%**	94.2%
	CHMR-HS without CTC (RDS+RAS+RES)	94.1%	85.4%	88.5%	91.4%
	CTC (RDS+RAS+RES)	90.6%	81.9%	84.6%	87.3%

**Table 3 sensors-22-08409-t003:** The comparison of recall (RE) for all motions and walking, running, and vertical-motion groups.

Data Type	Models	Recall			
		All Motions	Walking Group	Running Group	Vertical-Motion Group
**2D Features**	CHMR-HS (DS+AS+ES+RS)	92.2%	93.1%	92.4%	**94.2%**
	CHMR-HS without CTC (DS+AS+ES+RS)	91.2%	82%	835%	86.4%
	Zhao-I (DS)	90.6%	81%	77%	87.21%
	Zhao-II (DS)	90.1%	82%	81.5%	83%
	Shrestha (DS)	88.2%	78.2%	83%	82.9%
	CTC (DS+AS+ES+RS)	87.9%	80.3%	84.2%	83.2%
**3D Features**	CHMR-HS (RDS+RAS+RES)	**93.7%**	**94.4%**	**93%**	94.1%
	CHMR-HS without CTC (RDS+RAS+RES)	91.9%	83.9%	88.5%	90.8%
	CTC (RDS+RAS+RES)	78.9%	79.2%	82.1%	85.2%

**Table 4 sensors-22-08409-t004:** The comparison of precision (PR) for all motions and walking, running, and vertical-motion groups.

Data Type	Models	Precision			
		All Motions	Walking Group	Running Group	Vertical-Motion Group
**2D Features**	CHMR-HS (DS+AS+ES+RS)	81.9%	84.5%	82.9%	**84%**
	CHMR-HS without CTC (DS+AS+ES+RS)	81.1%	75.9%	76.2%	79.2%
	Zhao-I (DS)	81.8%	73%	70%	77.2%
	Zhao-II (DS)	79.1%	74%	72%	74%
	Shrestha (DS)	77.1%	80.3%	74%	72%
	CTC (DS+AS+ES+RS)	78.5%	73.9%	74.3%	74.2%
**3D Features**	CHMR-HS (RDS+RAS+RES)	**84.8%**	**86%**	**84%**	83.2%
	CHMR-HS without CTC (RDS+RAS+RES)	82.8%	73.1%	79%	82.3%
	CTC (RDS+RAS+RES)	81.4%	70.1%	73.2%	77.9%

**Table 5 sensors-22-08409-t005:** The comparison of F1-score (F1) for all motions and walking, running, and vertical-motion groups.

Data Type	Models	F1-Score			
		All Motions	Walking Group	Running Group	Vertical-Motion Group
**2D Features**	CHMR-HS (DS+AS+ES+RS)	86.7%	88.5%	87.4%	**88.8%**
	CHMR-HS without CTC (DS+AS+ES+RS)	85.8%	80.1%	81.2%	83.6%
	Zhao-I (DS)	85.9%	76.7%	73.3%	81.9%
	Zhao-II (DS)	84.2%	77.8%	76.4%	78.24%
	Shrestha (DS)	82.2%	79.2%	78.2%	77%
	CTC (DS+AS+ES+RS)	82.9%	75.2%	81.5%	88.7%
**3D Features**	CHMR-HS (RDS+RAS+RES)	**89%**	**90%**	**88.2%**	88.3%
	CHMR-HS without CTC (RDS+RAS+RES)	87.1%	78.3%	83.4%	86.3%
	CTC (RDS+RAS+RES)	83.5%	75.3%	86.9%	80.1%

**Table 6 sensors-22-08409-t006:** The performance comparison of all motions and walking, running, and vertical-motion groups, if using all of the data of the training dataset.

Data Type	Models	All Motions				Walking Group				Running Group				Vertical-Motion Group			
		Accuracy	Recall	Precision	F1-Score	Accuracy	Recall	Precision	F1-Score	Accuracy	Recall	Precision	F1-Score	Accuracy	Recall	Precision	F1-Score
**2D Features**	CHMR-HS	94.5%	92.2%	81.9%	86.7%	94.6%	93.1%	84.5%	88.5%	92.9%	92.4%	82.9%	87.4%	**95%**	**94.2%**	**84%**	**88.8%**
	CHMR-HS without CTC	93.4%	91.2%	81.1%	85.8%	84.5%	82%	75.9%	80.1%	86.1%	83.5%	76.2%	81.2%	89.3%	86.4%	79.2%	83.6%
	Zhao-I	93.1%	90.6%	81.8%	85.9%	82.55%	81%	73%	76.7%	79.2%	77%	70%	73.3%	88.25%	87.21%	77.2%	81.9%
	Zhao-II	92.7%	90.1%	79.1%	84.2%	83.34%	82%	74%	77.8%	81.9%	81.5%	72%	76.4%	83.37%	83%	74%	78.24%
	Shrestha	92.1%	88.2%	77.1%	82.2%	79.3%	78.2%	80.3%	79.2%	83.33%	83%	74%	78.2%	83.56%	82.9%	72%	77%
	CTC	91.3%	87.9%	78.5%	82.9%	81.6%	80.3%	73.9%	75.2%	83.5%	84.2%	74.3%	81.5%	86.2%	83.2%	74.2%	88.7%
**3D Features**	CHMR-HS	**95.2%**	**93.7%**	**84.8%**	**89%**	**95.3%**	**94.4%**	**86%**	**90%**	**93.8%**	**93%**	**84%**	**88.2%**	94.2%	94.1%	83.2%	88.3%
	CHMR-HS without CTC	94.1%	91.9%	82.8%	87.1%	85.4%	83.9%	73.1%	78.3%	88.5%	88.5%	79%	83.4%	91.4%	90.8%	82.3%	86.3%
	CTC	90.6%	78.9%	81.4%	83.5%	81.9%	79.2%	70.1%	75.3%	84.6%	82.1%	73.2%	86.9%	87.3%	85.2%	77.9%	80.1%

**Table 7 sensors-22-08409-t007:** The performance comparison of all motions and walking, running, and vertical-motion groups, if using all data of the training dataset.

Data Type	Models	All Motions															
		(2:1:1:1)				(1:2:1:1)				(1:1:2:1)				(1:1:1:2)			
		Accuracy	Recall	Precision	F1-Score	Accuracy	Recall	Precision	F1-Score	Accuracy	Recall	Precision	F1-Score	Accuracy	Recall	Precision	F1-Score
**2D Features**	CHMR-HS	**94.1%**	**93.2%**	**82.3%**	**85.2%**	**93.7%**	**93%**	**84.2%**	**87.3%**	90.2%	88.9%	80.1%	84.3%	89.3%	88.1%	79.1%	85.8%
	CHMR-HS without CTC	92.8%	90.8%	87.5%	84.6%	83.3%	81.1%	74.7%	79.3%	85.1%	86.1%	78.1%	82.6%	88.6%	84.1%	78.1%	84.2%
	Zhao-I	92.7%	90.1%	80.7%	83.9%	81.1%	79.5%	71.7%	73.5%	78.1%	75.9%	69.8%	71.3%	86.9%	86.1%	75.8%	80%
	Zhao-II	91.6%	89.1%	77.6%	82.4%	81.3%	80.8%	72.8%	75.6%	80.1%	79.6%	70.8%	74.6%	81.87%	81.8%	72.9%	76.5%
	Shrestha	91.8%	86.3%	75.8%	81.7%	77.5%	76.4%	79.6%	78.4%	81.9%	82.8%	72.5%	75.7%	81.8%	80.9%	70.7%	75%
	CTC	90.7%	86.5%	76.9%	81.8%	80.4%	79.2%	72.4%	74.3%	82.7%	83.7%	72.4%	80.7%	85.7%	82.7%	73.4%	81.9%
**3D Features**	CHMR-HS	93.4%	92.7%	80.8%	83.9%	92.3%	92.7%	81.5%	85.4%	**92.9%**	**89.7%**	**82.3%**	**85.7%**	**92.7%**	**89.9%**	**81.2%**	**86.1%**
	CHMR-HS without CTC	91.9%	90.1%	85.9%	83.1%	83%	80.2%	73.1%	78.3%	88.5%	87.9%	80.1%	84.1%	89.9%	85%	80%	86.2%
	CTC	90.3%	84.2%	74.2%	80.3%	79.2%	77.8%	71%	72.9%	83%	84%	73.8%	81.8%	86.4%	83.8%	75.3%	82.7%

**Table 8 sensors-22-08409-t008:** The performance comparisons of walking group under variable-time ratios = (2:1:1:1), (1:2:1:1), (1:1:2:1), and (1:1:1:2).

Data Type	Models	Walking Group															
		(2:1:1:1)				(1:2:1:1)				(1:1:2:1)				(1:1:1:2)			
		Accuracy	Recall	Precision	F1-Score	Accuracy	Recall	Precision	F1-Score	Accuracy	Recall	Precision	F1-Score	Accuracy	Recall	Precision	F1-Score
**2D Features**	CHMR-HS	94.8%	93.2%	85.5%	88.7%	94.8%	93.5%	85%	89.2%	93.5%	91.5%	82.1%	86.7%	90.1%	89.7%	81.3%	86.3%
	CHMR-HS without CTC	86.4%	83.6%	78.1%	82.5%	85.7%	84.1%	77.1%	82.3%	82.5%	80.6%	73.1%	79.2%	83.2%	80.7%	73%	78.1%
	Zhao-I	83.9%	83%	75.1%	78.1%	83.6%	82.4%	74.2%	78%	81%	79.2%	71.9%	75.3%	81%	79.3%	71.3%	74.3%
	Zhao-II	84.2%	83.5%	76.1%	79.1%	84.2%	81.2%	75.3%	79.1%	82.3%	80.1%	71.8%	75.9%	81.2%	80.3%	72.5%	75.2%
	Shrestha	80.1%	79.1%	81.3%	80.3%	81.2%	80.1%	82.1%	81.3%	78.1%	76.9%	79.3%	78.1%	77.9%	76.9%	78.6%	78.1%
	CTC	82.4%	81.4%	75.1%	76.3%	82.7%	81.4%	74.5%	77.1%	80.1%	79.2%	70.9%	73.8%	80.4%	78.6%	71.6%	73.5%
**3D Features**	CHMR-HS	**95.8%**	**94.8%**	**87.3%**	**91.3%**	**95.6%**	**94.8%**	**87.3%**	**92%**	**94.9%**	**93.8%**	**85.3%**	**89.3%**	**94.2%**	**84.9%**	**85.3%**	**88.9%**
	CHMR-HS without CTC	86.2%	84.3%	75.2%	79.3%	86.3%	85.1%	75%	79.2%	83.8%	83.1%	71.8%	77.2%	83.9%	82.8%	71.9%	76.8%
	CTC	82.5%	79.8%	72%	78.3%	82.1%	80.3%	71%	76.3%	80.2%	77.6%	69.3%	73.8%	80.2%	78.1%	68.9%	73.5%

**Table 9 sensors-22-08409-t009:** The performance comparisons of running group under variable-time ratios = (2:1:1:1), (1:2:1:1), (1:1:2:1), and (1:1:1:2).

Data Type	Models	Running Group															
		(2:1:1:1)				(1:2:1:1)				(1:1:2:1)				(1:1:1:2)			
		Accuracy	Recall	Precision	F1-Score	Accuracy	Recall	Precision	F1-Score	Accuracy	Recall	Precision	F1-Score	Accuracy	Recall	Precision	F1-Score
**2D Features**	CHMR-HS	93.5%	92.8%	83.5%	88.4%	93.4%	93.1%	83.5%	88.3%	92.1%	91.4%	81.2%	88.3%	91.5%	91.3%	81.2%	85.6%
	CHMR-HS without CTC	87.1%	84.8%	78.2%	82.4%	87.3%	84.3%	78.1%	82.3%	85.1%	82.3%	75.1%	80.1%	85.2%	82.1%	75.2%	80.2%
	Zhao-I	80.2%	78.3%	72.1%	75.1%	81.5%	78.3%	71.5%	74.3%	78.2%	75.8%	68%	72.5%	77.8%	75.4%	68.9%	71.8%
	Zhao-II	82.5%	83.4%	73.8%	78.1%	82.8%	83.4%	74.1%	78.2%	80.7%	79.8%	71.6%	74.5%	80.8%	80.5%	71.3%	75.3%
	Shrestha	84.5%	84.3%	75.4%	79.6%	85.1%	84.5%	76.2%	79.3%	82.5%	81%	72.9%	76.3%	82.1%	81.5%	72.6%	77.1%
	CTC	84.7%	85.3%	76.4%	82.6%	84.7%	85.7%	75.8%	82.6%	82.7%	83.6%	73.2%	80.5%	81.9%	82.7%	72.9%	80.1%
**3D Features**	CHMR-HS	**94%**	**93.5%**	**84.5%**	**89.7%**	**94.2%**	**93.6%**	**85.1%**	**89.7%**	**93.2%**	**92.1%**	**82.6%**	**86.7%**	**93.1%**	**84.2%**	**83.4%**	**86.7%**
	CHMR-HS without CTC	89.7%	89.2%	80%	84.6%	89.2%	89%	80%	84.2%	87.1%	88.2%	78.1%	82%	86.7%	87.2%	78.1%	85.1%
	CTC	85.1%	83.2%	74.6%	88.1%	85.3%	83.6%	74.8%	88.2%	83.5%	81.5%	71.5%	85.1%	82.8%	81.4%	72.1%	85.3%

**Table 10 sensors-22-08409-t010:** The performance comparisons of vertical-motion group under variable-time ratios = (2:1:1:1), (1:2:1:1), (1:1:2:1), and (1:1:1:2).

Data Type	Models	Vertical-Motion Group															
		(2:1:1:1)				(1:2:1:1)				(1:1:2:1)				(1:1:1:2)			
		Accuracy	Recall	Precision	F1-Score	Accuracy	Recall	Precision	F1-Score	Accuracy	Recall	Precision	F1-Score	Accuracy	Recall	Precision	F1-Score
**2D Features**	CHMR-HS	**94.3%**	**93.1%**	**82.1%**	**86.8%**	**94.1%**	**93.5%**	**82.4%**	**87.3%**	**95.3%**	**94.5%**	**84.7%**	**89.3%**	**95.4%**	**94.7%**	**86.1%**	**89.4%**
	CHMR-HS without CTC	88.6%	86%	78.7%	82.4%	88.1%	85.4%	78.1%	82.1%	90%	87.3%	80%	84%	90.4%	87%	80.4%	85.1%
	Zhao-I	87.1%	86.3%	76.1%	80%	86.8%	85.4%	76.1%	80.1%	89%	88.6%	78.3%	82%	89.1%	89.1%	78.3%	82.5%
	Zhao-II	81.5%	82.4%	72.6%	77.3%	82.4%	81.6%	72.6%	77.5%	84.6%	85.2%	75.1%	79.3%	85.1%	83.2%	74.6%	79.1%
	Shrestha	83.1%	81.5%	71.2%	76.1%	82.1%	81.8%	70.1%	75.6%	84.8%	84.1%	72.5%	78.1%	85.1%	84.1%	73.1%	79.2%
	CTC	85.1%	84.1%	73.2%	87.2%	85.7%	82.1%	73.2%	86.4%	88.2%	84.3%	75.2%	89.2%	89%	84.5%	76.1%	89.7%
**3D Features**	CHMR-HS	93.8%	93.5%	82.1%	87.3%	93.3%	93.2%	81.8%	87.2%	94.5%	94.4%	84.6%	89.2%	94.6%	95%	84.2%	89.9%
	CHMR-HS without CTC	91%	89.9%	81.3%	85.1%	90.4%	89.1%	81.2%	85.1%	92%	91.5%	83.5%	87.6%	92.3%	91.2%	84%	87.8%
	CTC	86.2%	84.1%	76.2%	79.2%	86.2%	84.2%	75.9%	79.1%	88.6%	86.8%	78.4%	82.1%	89.1%	86.3%	78.1%	82.1%

## Data Availability

Data sharing is not applicable to this article.

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
