# Peer review of "Multi-Feature Transformer-Based Learning for Continuous Human Motion Recognition with High Similarity Using mmWave FMCW Radar"

_sensors, 2022, doi:10.3390/s22218409_

Round 1
Reviewer 1 Report
This article proposed a method for effectively recognizing the continuous human motion recognition with high similarity. The result shows it works well. The article description is detailed and logically clear.
1. On line 24, there is a case error in ", For instance,". The multiplication symbols in function (1),(2) and (3) should be consistent.
2. In section 3.2, why obtaining the best solution of Y means that eachl is the best solution? As far as I concerned, getting the overall optimal solution (Y) does not mean that the optimal solution can be obtained for a single entity (l). If this were the case, this might mean that the recognition rate is 100%, but in fact it is not. Please explain.
3. Though the raw data preprocessing module has been used for background removal operation. I still want to know if the data set is collected in different background environment and if the algorithm works well when the background is different, especially when the background is crowded, not as empty as shown in Figure 9.
4. The experiment requires consideration of how to handle multiple moving targets within the detection range. Does the algorithm still work well when there are moving objects in the background or when multiple people are moving in the detection range?
Author Response
This article proposed a method for effectively recognizing the continuous human motion recognition with high similarity. The result shows it works well. The article description is detailed and logically clear.
Answer: Thanks for the carefully reading and fully understanding.
- On line 24, there is a case error in ", For instance,". The multiplication symbols in function (1),(2) and (3) should be consistent.
Answer: Thanks for carefully checking. The error of line 24 is corrected. In addition, the multiplication symbols in function (1), (2), (3), and (4) are consistent in the revised paper. - In section 3.2, why obtaining the best solution of Y means that each is the best solution? As far as I concerned, getting the overall optimal solution (Y) does not mean that the optimal solution can be obtained for a single entity (l). If this were the case, this might mean that the recognition rate is 100%, but in fact it is not. Please explain.
Answer: Sorry for the misunderstanding due to our un-clear writing, the best solutions will be selected from all possible training samples I, all possible motions J, and all possible motion categories K, such that we have , of our loss functions of our three decoders in equations (1), (2), and (3), respectively. The overall optimal solution is determined by equation (4). In the Transformer-based schemes, it is not easily to achieve the 100% of recognition rate. However, our proposed scheme can obtain the better result than that of existing published CHMR methods, under the same selected environment. However, we re-organize Section 3.2. - Though the raw data preprocessing module has been used for background removal operation. I still want to know if the data set is collected in different background environment and if the algorithm works well when the background is different, especially when the background is crowded, not as empty as shown in Figure 9.
Answer: Thanks for very in-sight point. In fact, we have considered this issue, so we have collected data from two different rooms with the different environments. We collected 80% raw data from a big-space room without the complicated background, and we also collected 20% raw data from the another small-space room with the more complicated background. We added these data collections in the revised paper.
To avoid the different background issue, we adopt the MTI (Moving Target Indication) filter algorithm in our paper. The signal of the non-moving object (no phase difference) will be removed to ensure that the detected object is only detected in a moving condition, such that the movement, or called as phase difference, can be directly retained. We may also called this as the static removal operation. - The experiment requires consideration of how to handle multiple moving targets within the detection range. Does the algorithm still work well when there are moving objects in the background or when multiple people are moving in the detection range ?
Answer: Thanks for in-sight suggestion. However, we are very sorry that the research scope of continuous human motion recognition (CHMR) work is only for the single moving object, including the radar samples. Unfortunately, the continuous human motion recognition for multiple moving targets will be future work under designing more in-depth algorithm and different radar samples with multiple moving targets. We add these parts into the future work of Conclusion.

Reviewer 2 Report
Authors presented "Multi-Feature Transformer-Based Learning for Continuous Human Motion Recognition with High Similarity using mmWave FMCW Radar"
The work is very interesting and sounds quality for publication. few comments are listed belwo.
1. What is the criteria to design FMCW Radar
2. Is background denoising removal sufficient to provide accurate results? What will happen if minute noise is present in removal?
3. How it is differentthat other RADAR techniques?
4. Can you add a comparison table of radar technique applications ofr the same work?
5. few typos are present, check carefully.
Author Response
Authors presented "Multi-Feature Transformer-Based Learning for Continuous Human Motion Recognition with High Similarity using mmWave FMCW Radar"The work is very interesting and sounds quality for publication. few comments are listed below.
Answer: Thanks for fully understanding our contribution. We try to revise our paper to provide a more high-quality paper for the publication in Sensors.
- What is the criteria to design FMCW Radar.
Answer: The criteria to design FMCW Radar is that FMCW Radars transmit frequency modulated continuous wave signal and reflected FMCW signal from the remote target is mixed with the transmitted signal to generate a signal at an intermediate frequency (IF), which is used in the range, Doppler, and angle calculation. We added these explanations in the Introduction. - Is background denoising removal sufficient to provide accurate results? What will happen if minute noise is present in removal ?
Answer: Thanks for the carefully comments. The purpose of the background denoising algorithm is to extract the phase frequency between the signals. In our work, we only take the data frames with the largest signal after the Fourier transformation and then concatenate these data frames together. The intuition is a continuous echo map of the torso. If one or two noises are larger than the torso reflections or there is a small noise in the process of denoising, it doesn't matter for misguided training, since one or two noises of the tens to hundreds of data frames will be moderately retained and will not be particularly problematic. We add these explanations into the background denoising removal parts. - How it is different that other RADAR techniques?
Answer: The key advantage of mmWave FMCW radar, compared with LiDar, is not affected by light and low cost. Due to these characteristic, it is very useful to apply mmWave FMCW radar applications for automatic car driving and home care, especially for the night care for the elderly. We added these explanations in the Introduction. - Can you add a comparison table of radar technique applications of the same work?
Answer: Thanks for very useful suggestion, we add a new subsection 5.3.5 to discuss with the high similarity by adding five Tables to further show our performance achievements by using the mmWave FMCW radars. In the revised paper, we also provide the detailed algorithms to provide the further readers.
few typos are present, check carefully.
Answer: Thanks for the carefully checking, we have carefully read and check this paper, to correct the possible typos.
